# HIGH-PROBABILITY BOUNDS FOR THE LAST ITERATE OF CLIPPED SGD

**Savelii Chezhegov**
BRAIn Lab*, MIRAI[†],
Sber AI Lab

**Daniela A. Parletta**
University of Genoa

**Andrea Paudice**
Aarhus University

**Eduard Gorbunov**
MBZUAI[‡]

## ABSTRACT

We study the problem of minimizing a convex objective when only noisy gradient estimates are available. Assuming that stochastic gradients have finite $\alpha$-th moments for some $\alpha \in (1, 2]$, we establish – for the first time – a high-probability convergence guarantee for the last iterate of clipped stochastic gradient descent (Clipped-SGD) on smooth objectives. In particular, we prove a rate of $1/K^{(2\alpha-2)/(3\alpha)}$ with only polylogarithmic dependence on the confidence parameter. In addition, we introduce a new technique for deriving in-expectation convergence guarantees from high-probability bounds for methods with almost surely bounded updates, and apply it to obtain expectation guarantees for Clipped-SGD. Finally, we complement our theoretical analysis with empirical results that support and illustrate our findings.

## 1 INTRODUCTION

Stochastic first-order optimization methods such as SGD (Robbins & Monro, 1951), Adam (Kingma & Ba, 2014), and their numerous variants are central to the training of modern machine learning models. In practice, these algorithms are almost always combined with additional techniques that enhance stability and performance. One such technique – *gradient clipping* (Pascanu et al., 2013) – has become a standard component in the training of large language models (LLMs) (Devlin et al., 2019; Brown et al., 2020; Fedus et al., 2022; Touvron et al., 2023).

Originally introduced to address exploding gradients in recurrent neural networks, gradient clipping has since proven valuable well beyond this motivation. It has been shown to improve convergence under generalized smoothness conditions (Zhang et al., 2020a), to provide robustness against heavy-tailed noise where gradient variance can be unbounded (Zhang et al., 2020b), and to enable *strong high-probability* convergence guarantees (Gorbunov et al., 2020; Cutkosky & Mehta, 2021; Parletta et al., 2024; Sadiev et al., 2023; Nguyen et al., 2023). Nevertheless, for *convex problems*, most existing theoretical results for clipped methods (including Clipped-SGD) analyze *averaged iterates*, while the practically more relevant *last iterate* remains largely unexplored.

**Contributions.** In this work, we close a long-standing gap in the theory of clipped stochastic gradient descent by analyzing the *last iterate* under heavy-tailed noise. Our main contributions are:

- **First high-probability last-iterate guarantees for Clipped-SGD.** We establish the first high-probability convergence rate for the *last iterate* of Clipped-SGD on convex smooth objectives. Assuming stochastic gradients have finite $\alpha$-th moments with $\alpha \in (1, 2]$, we prove that after $K$ iterations the method achieves an error of at most

$$\mathcal{O}\left(\frac{\text{polylog}(K/\delta)}{K^{2(\alpha-1)/3\alpha}}\right)$$

with failure probability at most $\delta \in (0, 1)$, see Theorem 1 and Corollary 1. In the special case of $\alpha = 2$, this results in a polynomial gap compared to the best-known in-expectation

---

*Basic Research of Artificial Intelligence Laboratory

[†]Moscow Independent Research Institute of Artificial Intelligence

[‡]Mohamed bin Zayed University of Artificial Intelligence

rate of $1/\sqrt{K}$, where clipping is unnecessary. Crucially, our result covers the full spectrum of heavy-tailed noise and provides high-probability guarantees for a single run – significantly stronger than standard in-expectation bounds.

- **General analysis of step-size and clipping schedules.** We develop a unified analysis for polynomially decaying step-sizes and varying clipping levels, bounding the optimization error as a function of these schedules. This yields principled guidelines for tuning and identifies optimal exponents, while avoiding restrictive assumptions such as a bounded optimization domain.

- **Any-time parameter choices.** Our parameter selection is *horizon-free*: it does not require prior knowledge of the number of iterations and remains valid in streaming or indefinite-training scenarios, where restarting schemes are impractical. Our results hold without reliance on large minibatches, making them applicable in resource-constrained settings.

- **From high-probability to in-expectation convergence.** We develop a general technique for converting high-probability convergence guarantees into in-expectation convergence rates. The approach applies to any method for which the convergence criterion (e.g., the function sub-optimality of the last iterate) is almost surely polynomially bounded in terms of the total number of iterations $K$. In particular, our technique can be applied to any high-probability result established for algorithms with almost surely bounded updates, including Clipped-SGD, normalized SGD with momentum (Cutkosky & Mehta, 2020), SignSGD (Bernstein et al., 2018), Muon (Jordan et al., 2024), and many other related methods. In Section 4.2, we demonstrate this approach on the example of Clipped-SGD.

We complement our result with empirical evidence supporting the advantage of the last iterate over the average.

## 2 NOTATION AND PROBLEM SETUP

In this section, we introduce the main notation and discuss the assumptions used in the analysis.

**Notation.** The norm $\|x\| := \sqrt{\langle x, x \rangle}$ denotes the Euclidean norm in $\mathbb{R}^d$. $\mathbb{E}_\xi[\,\cdot\,]$ denotes the expectation w.r.t. the random variable $\xi$. We also denote the clipping operator as $\mathrm{clip}(x, \lambda) := \min\left\{1, \frac{\lambda}{\|x\|}\right\} x$. The initial distance, i.e., the distance between the starting point $x_0$ and a solution[1] $x^*$, we denote as $R_0 := \|x_0 - x^*\|$.

**Problem.** We study the following problem:

$$\min_{x \in \mathbb{R}^d} f(x), \tag{1}$$

under the following standard hypothesis.

**Assumption 1** (Convexity). *The differentiable function $f$ is convex, i.e.,*

$$f(y) \geq f(x) + \langle \nabla f(x), y - x \rangle \quad \forall x, y \in \mathbb{R}^d.$$

Moreover, we also assume that there exists optimal point $x^*$.

**Assumption 2.** *There exist an optimal point $x^* \in \mathbb{R}^d$ such that for $x \in \mathbb{R}^d$*

$$f(x) \geq f(x^*) > -\infty.$$

In addition to convexity, we assume that $f$ is $L$-smooth.

**Assumption 3** (Smoothness). *The differentiable function $f$ is $L$-smooth, i.e.,*

$$f(y) \leq f(x) + \langle \nabla f(x), y - x \rangle + \frac{L}{2}\|y - x\|^2 \quad \forall x, y \in \mathbb{R}^d.$$

Finally, although the optimizer does not have direct access to $f$, we assume access to a stochastic gradient oracle $\nabla f(x, \xi)$ satisfying the following condition.

---

[1]Our results hold for any solution of the considered problem.

**Assumption 4** (Stochastic oracle). *The stochastic oracle $\nabla f(x, \xi)$ is unbiased and has bounded $\alpha$-th central moment with $\alpha \in (1, 2]$, i.e.*

$$\mathbb{E}\left[\nabla f(x, \xi)\right] = \nabla f(x); \qquad \mathbb{E}\left[\|\nabla f(x, \xi) - \nabla f(x)\|^{\alpha}\right] \leq \sigma^{\alpha}.$$

This assumption was introduced by Nemirovskij & Yudin (1983) and later rediscovered by Zhang et al. (2020b), after which it has become standard in the analysis of stochastic methods under heavy-tailed noise. For problem (1), we study Clipped-SGD with time-varying stepsize $\gamma_k$ and clipping level $\lambda_k$:

$$x_{k+1} = x_k - \gamma_k \cdot \text{clip}(\nabla f_{\xi_k}(x_k), \lambda_k),$$

where $\nabla f_{\xi_k}(x_k) = \nabla f(x_k, \xi_k)$ is the stochastic gradient sampled independently of the past.

**Stochastic optimization.** The above problem encompasses, as a special case, the classical stochastic optimization problem:

$$\min_{x \in \mathbb{R}^d} \left[ f(x) := \mathbb{E}_{\xi \sim \mathcal{D}}\left[\ell(x, \xi)\right] \right], \tag{2}$$

where $\ell$ is the loss function, $x$ are the model parameters and $\xi$ represents the randomness due to data sampling from the unknown distribution $\mathcal{D}$. In this setting, the stochastic gradient can be computed from the sample $\xi$ as $\nabla f(x, \xi) = \nabla \ell(x, \xi)$. Note that, when $\xi = (Z, Y) \in \mathbb{R}^d \times \mathbb{R}$, Equation (2) also covers the statical supervised learning problem. Another important special case of Equation (2) is the finite-sum setting $f(x) = \sum_{i=1}^{n} f_i(x)$, which underlies many training procedures in machine learning.

**High-probability convergence guarantees.** For an iterative sequence $\{x_k\}_{k=0}^{K}$ (produced by some stochastic method) and a target criterion $C(\{x_k\}_{k=0}^{K})$, the standard goal is to ensure

$$\mathbb{E}\left[C\left(\{x_k\}_{k=0}^{K}\right)\right] \leq \varepsilon.$$

Such in-expectation bounds describe average behavior, but they do not capture the variability of the random process. *High-probability bounds*, by contrast, guarantee that the performance criterion is met with confidence at least $1 - \delta$, i.e.,

$$\mathbb{P}\left\{C\left(\{x_k\}_{k=0}^{K}\right) \leq \varepsilon\right\} \geq 1 - \delta.$$

thereby directly limiting the probability of unfavorable deviations.

Although one can obtain such bounds from expectation guarantees via Markov's inequality, this typically yields rates with an undesirable polynomial $1/\delta$ dependence. Modern approaches instead achieve bounds with only *polylogarithmic* dependence on $1/\delta$, which greatly improves reliability and reduces the number of iterations required to reach a target confidence level. In other words, the goal of high-probability convergence analysis is to establish convergence rates of the same order as the optimal in-expectation guarantees, with only a minimal dependence on the confidence parameter $\delta$, ideally $\mathcal{O}(\sqrt{\log(1/\delta)})$.

## 3 RELATED WORKS

The literature on SGD and Clipped-SGD is vast and multifaceted, and a comprehensive survey is beyond the scope of this work. In what follows, we focus only on the closely related works.

**In-expectation convergence bounds.** Early studies on SGD for smooth and non-smooth (but Lipschitz) objectives investigated convergence in expectation under finite-variance noise (Nemirovskij & Yudin, 1983; Nemirovski et al., 2009; Ghadimi & Lan, 2013a;b). In this setting, the average iterate of SGD achieves a rate of $\mathcal{O}(1/\sqrt{K})$, which is known to be optimal (Agarwal et al., 2012). Similar optimal rates for the last iterate in the non-smooth case were established in (Shamir & Zhang, 2013; Jain et al., 2021), while for smooth objectives the best known rate remained $\mathcal{O}(1/K^{1/3})$ (Moulines & Bach, 2011) for a long time, until (Liu & Zhou, 2024) — building on ideas from (Zamani & Glineur, 2023) — proved the optimal rate $\mathcal{O}(1/\sqrt{K})$, thereby unifying the analysis of both smooth and non-smooth cases.

Under the more general bounded $\alpha \in (1, 2]$ moments model considered in this work, Zhang et al. (2020b) show that plain SGD does not converge (in terms[2] of $\mathbb{E}[\|x_k - x^*\|^2]$), and prove in-expectation convergence bounds for Clipped-SGD for non-convex smooth and strongly convex problems with bounded gradients. Vural et al. (2022) derive average-iterate $\mathcal{O}(1/K^{(\alpha-1)/\alpha})$ bound for Stochastic Mirror Descent (SMD) over convex Lipschitz objectives and show that it is optimal. In the same non-smooth setting, Parletta et al. (2025) show that the last iterate of Clipped-SGD enjoys the same optimal rate. Moreover, for strongly convex objectives, Jakovetić et al. (2023) investigate the last-iterate convergence of a general class of robust SGD variants assuming only $\alpha = 1$. However, these results require additional assumptions, such as symmetry of the noise distribution and constraints on its *effective dimension*. To the best of our knowledge, there exist no *last-iterate* in-expectation convergence bounds for Clipped-SGD in the case of convex $L$-smooth problems with the stochastic oracle satisfying Assumption 4.

**High-probability convergence bounds.** The first high-probability results for SGD were established under sub-Gaussian noise assumptions (Nemirovski et al., 2009; Harvey et al., 2019b), which are considerably stronger than those considered in this work. Moreover, these guarantees apply only to the average iterate. In this setting and for non-smooth objectives, Liu et al. (2023) showed that the average iterate of Stochastic Mirror Descent (and hence of SGD) converges at the optimal rate with only a modest confidence overhead of $\mathcal{O}(\sqrt{\log(1/\delta)})$. Similar last-iterate guarantees (including for smooth objectives) were later obtained by Eldowa & Paudice (2024) and Liu & Zhou (2024). Both works also relaxed the sub-Gaussian assumption to the broader class of sub-Weibull tails (Vladimirova et al., 2020), which were further explored in non-convex settings by Madden et al. (2024). While more general, these tail models still imply the existence of moments of all orders.

In contrast, under the heavy-tailed noise model studied here, several works (Nazin et al., 2019; Gorbunov et al., 2020; Nguyen et al., 2023; Sadiev et al., 2023; Gorbunov et al., 2024; Parletta et al., 2024; 2025) have shown that clipping yield convergence of the average iterate at rate $\widetilde{\mathcal{O}}(1/K^{(\alpha-1)/\alpha})$[3] for convex $L$-smooth objectives. To the best of our knowledge, results for the last iterate in this regime require additional structural assumptions such as strong convexity or the PL condition (Sadiev et al., 2023). Our work closes this gap by establishing, for the first time, high-probability convergence rates for the last iterate of SGD on convex smooth objectives. Finally, refinements of the $\alpha \in (1, 2]$ model have been proposed in Puchkin et al. (2024), who demonstrate that in certain cases it is possible to surpass the $\widetilde{\mathcal{O}}(1/K^{(\alpha-1)/\alpha})$ barrier.

## 4 MAIN RESULTS

In this section, we state our high-probability last-iterate guarantee for Clipped-SGD and explain the ideas behind its proof.

**Theorem 1.** *Suppose that Assumptions 1, 2, 3 and 4 hold. Then, if we choose*

$$\gamma_k = \min \left\{ \frac{1}{320L \ln\left(\frac{6(k+1)^2}{\delta}\right)}, \frac{B}{80 \cdot 13^{1/\alpha}(k+1)^\beta \ln\left(\frac{6(k+1)^2}{\delta}\right)} \right\}, \tag{3}$$

$$\lambda_k = \frac{R_0 \min\left\{\frac{1}{\sqrt{d_k}}, 1\right\}}{80\gamma_k \ln^{1/2}\left(\frac{6(k+1)^2}{\delta}\right)}, \tag{4}$$

---

[2]In the case of convex non-smooth problems, i.e., assuming that $\mathbb{E}[\|\nabla f(x, \xi)\|^\alpha \leq G^\alpha]$, Fatkhullin et al. (2025) prove $\mathcal{O}(1/K^{(\alpha-1)/\alpha})$ average-iterate convergence rate of SGD. The same result was independently shown by Liu (2025) for the average and *last* iterates of SGD (a.k.a. Online Gradient Descent), Dual Averaging (Nesterov, 2009), and AdaGrad Streeter & McMahan (2010); Duchi et al. (2011). Moreover, Liu (2025) also derive similar results for the average iterates of SGD, Dual Averaging, and AdaGrad for convex smooth objectives. Importantly, both papers (Fatkhullin et al., 2025; Liu, 2025) assume boundedness of the domain.

[3]The $\widetilde{\mathcal{O}}$ hides poly-logarithmic factors. Nazin et al. (2019); Gorbunov et al. (2020; 2024); Parletta et al. (2024) consider the case of $\alpha = 2$, which is often considered as heavy-tailed noise in the literature since the majority of prior high-probability results relied on sub-Gaussian noise assumption.

Table 1: Comparison of the state-of-the-art *non-accelerated* in-expectation and high-probability convergence results for SGD/Clipped-SGD-like methods applied to *smooth convex* problems.

| Reference | Convergence type | Iterate | Stochasticity | Rate |
|---|---|---|---|---|
| (Ghadimi & Lan, 2013b) | In-expectation | Average | As. 4, $\alpha = 2$ | $\mathcal{O}\left(\frac{LR_0^2}{K} + \frac{R_0\sigma}{\sqrt{K}}\right)$ |
| (Taylor & Bach, 2019) | In-expectation | Last | As. 4, $\alpha = 2$ | $\mathcal{O}\left(\frac{LR_0^2 + \sigma^2}{K^{1/3}}\right)$ |
| (Liu & Zhou, 2024) | In-expectation | Last | As. 4, $\alpha \in (1, 2]$ | $\mathcal{O}\left(\frac{LR_0^2}{K^{2(\alpha-1)/\alpha}} + \frac{R_0\sigma}{K^{(\alpha-1)/\alpha}}\right)$ |
| (Ghadimi & Lan, 2013b) | High probability | Average | Sub-Gaussian | $\mathcal{O}\left(\frac{LR_0^2}{K} + \frac{R_0\sigma}{\sqrt{K}}\right)$ |
| (Liu & Zhou, 2024) | High probability | Last | Sub-Weibull | $\mathcal{O}\left(\frac{LR_0^2}{K} + \frac{R_0\sigma}{\sqrt{K}}\right)$ |
| (Nguyen et al., 2023) | High probability | Average | As. 4, $\alpha \in (1, 2]$ | $\widetilde{\mathcal{O}}\left(\frac{LR_0^2}{K} + \frac{R_0\sigma}{K^{(\alpha-1)/\alpha}}\right)^{(1)}$ |
| **This work** | High probability | Last | As. 4, $\alpha \in (1, 2]$ | $\widetilde{\mathcal{O}}\left(\frac{LR_0^2}{K} + \frac{D}{K^{2(\alpha-1)/3\alpha}}\right)^{(2)}$ |

[1] The rate from Nguyen et al. (2023) has better logarithmic factor than the one from Sadiev et al. (2023).
[2] $D := \max\left\{R_0\sigma, L^{(\alpha-1)/(3\alpha-1)}R_0^{(4\alpha-2)/(3\alpha-1)}\sigma^{2\alpha/(3\alpha-1)}, L^{1/3}R_0^{4/3}\sigma^{2/3}\right\}$.

*where $B$ is chosen as*

$$B = \min\left\{\frac{R_0}{\sigma}, \frac{R_0^{\frac{2\alpha}{3\alpha-1}}}{\sigma^{\frac{2\alpha}{3\alpha-1}}L^{\frac{\alpha-1}{3\alpha-1}}}, \frac{R_0^{\frac{2}{3}}}{\sigma^{\frac{2}{3}}L^{\frac{1}{3}}}\right\},$$

*and parameter $\beta$ satisfies*

$$\beta \geq \frac{2+\alpha}{3\alpha},$$

*then, after $K$ iterations of Clipped-SGD, we have that*

$$f(x_K) - f^*$$
$$= \mathcal{O}\left(\left(\frac{LR_0^2}{K} + \frac{\max\left\{R_0\sigma, L^{\frac{\alpha-1}{3\alpha-1}}R_0^{\frac{4\alpha-2}{3\alpha-1}}\sigma^{\frac{2\alpha}{3\alpha-1}}, L^{\frac{1}{3}}R_0^{\frac{4}{3}}\sigma^{\frac{2}{3}}\right\}}{K^{1-\beta}}\right)\ln^2\left(\frac{6(K+1)^2}{\delta}\right)\right)$$

*holds with probability at least $1 - \delta\sum_{t=1}^{K}\frac{1}{t^2}$.*

**Corollary 1.** Let the conditions of Theorem 1 hold. Then, if we choose $\beta$ in the optimal way, i.e. $\beta = \frac{2+\alpha}{3\alpha}$, we derive that

$$f(x_K) - f^* = \tilde{\mathcal{O}}\left(\frac{LR_0^2}{K} + \frac{\max\left\{R_0\sigma, L^{\frac{\alpha-1}{3\alpha-1}}R_0^{\frac{4\alpha-2}{3\alpha-1}}\sigma^{\frac{2\alpha}{3\alpha-1}}, L^{\frac{1}{3}}R_0^{\frac{4}{3}}\sigma^{\frac{2}{3}}\right\}}{K^{\frac{2\alpha-2}{3\alpha}}}\right)$$

holds with probability at least $1 - 2\delta$. Here $\tilde{\mathcal{O}}(\cdot)$ denotes polylogarithmic dependency.

*Proof.* It is enough to substitute the equation for $\beta$ in the result of Theorem 1 and clarify that

$$1 - \delta\sum_{t=1}^{K}\frac{1}{t^2} \geq 1 - \delta\sum_{t=1}^{\infty}\frac{1}{t^2} = 1 - \frac{\delta\pi^2}{6} \geq 1 - 2\delta.$$

This finishes the proof. $\qquad\square$

## 4.1 PROOF SKETCH AND TECHNICAL NOVELTIES

We now outline the analysis and highlight the three key innovations.

**Potential-based *high-probability* convergence proof.** We analyze the method using the following potential:

$$\Phi_k = d_k(f(x_k) - f^*) + L\|x_k - x^*\|^2, \quad d_{k+1} = d_k + 2\gamma_k L, \ d_0 = 0.$$

This potential coincides with the one proposed by Bansal & Gupta (2017) to study the convergence of Gradient Descent. Moreover, Taylor & Bach (2019) use $\Phi_k$ to derive last-iterate *in-expectation* convergence rates (see Table 1). Since we focus on high-probability convergence, our descent lemma differs from that of Taylor & Bach (2019) and leads to

$$\Phi_K \leq \Phi_0 - \sum_{k=0}^{K-1} 2\gamma_k L \langle x_k - x^*, \theta_k \rangle - \sum_{k=0}^{K-1} d_k \gamma_k \langle \nabla f(x_k), \theta_k \rangle + \sum_{k=0}^{K-1} (d_{k+1} + 1)L\gamma_k^2 \|\theta_k\|^2, \quad (5)$$

where $\theta_k := g_k - \nabla f(x_k)$ and $g_k := \text{clip}(\nabla f_{\xi_k}(x_k), \lambda_k)$. The three martingale-type sums on the right-hand side are controlled via Bernstein/Freedman inequalities using a time-varying failure budget $\delta_t \sim 1/t^2$, followed by a union bound over $t$.

**Clipping level $\lambda_k \sim 1/\sqrt{d_k}$.** A key technical ingredient of our analysis is the choice

$$\lambda_k = \frac{R_0}{80\gamma_k \ln^{1/2}\left(\frac{6(k+1)^2}{\delta}\right)} \cdot \min\left\{\frac{1}{\sqrt{d_k}}, 1\right\}$$

rather than a scaling proportional to $1/\gamma_k$, as commonly used in existing average-iterate convergence results (e.g., (Sadiev et al., 2023; Nguyen et al., 2023)). The additional factor $\min\{1/\sqrt{d_k}, 1\}$ is motivated by the following observation. Since we prove by induction that $f(x_k) - f^*$ decreases as $\mathcal{O}(1/d_k)$ with high probability, it follows that $\|\nabla f(x_k)\| \leq \sqrt{2L(f(x_k) - f^*)} = \mathcal{O}(1/\sqrt{d_k})$. At the same time, the standard smoothness bound $\|\nabla f(x_k)\| \leq L\|x_k - x^*\|$ remains valid. This refined scaling of $\lambda_k$ therefore allows us to more effectively balance the bias and variance terms introduced by clipping under Assumption 4, leading to sharper high-probability bounds.

**Horizon-agnostic schedules and $\log$-factors.** Both $\gamma_k$ and $\lambda_k$ are any-time, i.e., no prior knowledge of $K$ is assumed. This influences two aspects of the proof:

- Similarly to the prior high-probability analysis of Clipped-SGD (Gorbunov et al., 2020; Sadiev et al., 2023) for smooth convex objectives, we bound the sums from (5) for each $K > 0$ with high probability and then apply the union bound for estimating the probability of the "good" event $E_K$. However, since the horizon is unknown, we cannot select the failure probability at each step as $\delta/K$ for each $k = 0, \ldots, K - 1$. Instead, the failure probability for step $k$ is upper bounded by $\delta/k^2$ in our proof. The choice $\delta_k \sim 1/k^2$ (and the resulting $\sum_k \delta_k \leq \delta$) introduces at most polylogarithmic dependence on $1/\delta$ in the final bound, while keeping the schedules horizon-free.

- Moreover, due to the horizon independence of the parameters, the derived upper bound for $\Phi_K$ contains a logarithmic factor $\sim \ln(6(K+1)^2/\delta)$. This leads to the additional exponent in the final bound (for comparison, see Sadiev et al. (2023) or Nguyen et al. (2023)). We refer to Appendix B.3 for further details.

- We also note that $\gamma_k$ and $\lambda_k$ depend on the failure probability $\delta$, which is a common practice in high-probability convergence results (Nazin et al., 2019; Gorbunov et al., 2020; Nguyen et al., 2023; Sadiev et al., 2023; Parletta et al., 2025). To the best of our knowledge, *almost*[4] all existing high-probability results with $\delta$-independent parameters choice are derived under either sub-Gaussian (Li & Orabona, 2020; Harvey et al., 2019a; Jain et al., 2021; Liu et al., 2023) or sub-Weibull (Madden et al., 2024) noise assumption. In both of these cases, the analysis does not require proving by induction that the iterates remain bounded with high probability due to the strengths of the assumptions. In contrast, our analysis does rely on inductive argument, which is the key reason why $\gamma_k$ and $\lambda_k$ are dependent on $\delta$ in our

---

[4]Hübler et al. (2025); Kornilov et al. (2025) derive high-probability convergence results for normalized SGD and SignSGD respectively for non-convex (generally) smooth objectives. The results are shown for the $\delta$-agnostic hyper-parameters' choices. However, both results rely on the usage of large mini-batches, which is often prohibitively expensive.

results. Nevertheless, we note that the dependence of $\gamma_k$ and $\lambda_k$ on $\delta$ does not artificially stabilize the dynamics of the method and our result does imply meaningful in-expectation bound as we explain in Section 4.2.

**Remark 1.** When $\alpha \to 1$, then our result shows convergence to some (finite) neighborhood, which is well-aligned with existing average-iterate results (see Table 1) and the lower bound for the Lipschitz convex case (Vural et al., 2022).

## 4.2 FROM HIGH-PROBABILITY TO IN-EXPECTATION CONVERGENCE

Since $\gamma_k$ and $\lambda_k$ are $\delta$-dependent, one cannot obtain an in-expectation guarantee from our high-probability result via integration of the tail bound. Nevertheless, one can derive a meaningful in-expectation convergence guarantee in another way. Indeed, for any fixed $\delta \in (0, 1)$ we can choose sequences of parameters $\{\gamma_k\}_{k=0}$ and $\{\lambda_k\}_{k=0}$ according to the Theorem 1. In this case, the following inequality holds with probability at least $1 - 2\delta$:

$$
f(x_K) - f^*
$$
$$
\leq \widetilde{C} \ln^2 \left( \frac{6(K+1)^2}{\delta} \right) \left( \frac{LR_0^2}{K} + \frac{\max\left\{ R_0 \sigma, L^{\frac{\alpha-1}{3\alpha-1}} R_0^{\frac{4\alpha-2}{3\alpha-1}} \sigma^{\frac{2\alpha}{3\alpha-1}}, L^{\frac{1}{3}} R_0^{\frac{4}{3}} \sigma^{\frac{2}{3}} \right\}}{K^{1-\beta}} \right), \quad (6)
$$

where $\widetilde{C}$ is some numerical constant. What is more, for Clipped-SGD we can also bound $f(x_K) - f^*$ with probability 1. To show that, let us bound the distance to $x^*$ at iteration $K$:

$$
\|x_K - x^*\| = \left\| x_0 - \sum_{k=0}^{K-1} \gamma_k \cdot \text{clip}(\nabla f_{\xi_k}(x_k), \lambda_k) - x^* \right\|
$$
$$
\leq \|x_0 - x^*\| + \sum_{k=0}^{K-1} \gamma_k \|\text{clip}(\nabla f_{\xi_k}(x_k), \lambda_k)\|
$$
$$
\leq \|x_0 - x^*\| + \sum_{k=0}^{K-1} \gamma_k \lambda_k \overset{(4)}{\leq} R_0 + \sum_{k=0}^{K-1} \frac{R_0}{80} \leq KR_0.
$$

Therefore, we can upper bound the function sub-optimality using Assumption 3:

$$
f(x_K) - f^* \leq \frac{L}{2} \|x_K - x^*\|^2 \leq \frac{LR_0^2 K^2}{2}. \quad (7)
$$

Next, since the random variable $f(x_K) - f^*$ is bounded as in (6) with probability at least $1 - 2\delta$, and is bounded as in (7) with probability 1, the expectation of $f(x_K) - f^*$ can be bounded as follows:

$$
\mathbb{E}[f(x_K) - f^*]
$$
$$
\leq (1 - 2\delta)\widetilde{C} \log^2 \left( \frac{6(K+1)^2}{\delta} \right) \cdot \left( \frac{LR_0^2}{K} + \frac{\max\left\{ R_0 \sigma, L^{\frac{\alpha-1}{3\alpha-1}} R_0^{\frac{4\alpha-2}{3\alpha-1}} \sigma^{\frac{2\alpha}{3\alpha-1}}, L^{\frac{1}{3}} R_0^{\frac{4}{3}} \sigma^{\frac{2}{3}} \right\}}{K^{1-\beta}} \right)
$$
$$
+ \delta LR_0^2 K^2
$$
$$
\leq (1 - 2\delta)\widetilde{C} \log^2 \left( \frac{6(K+1)^2}{\delta} \right) \cdot \left( \frac{LR_0^2}{K} + \frac{\max\left\{ R_0 \sigma, L^{\frac{\alpha-1}{3\alpha-1}} R_0^{\frac{4\alpha-2}{3\alpha-1}} \sigma^{\frac{2\alpha}{3\alpha-1}}, L^{\frac{1}{3}} R_0^{\frac{4}{3}} \sigma^{\frac{2}{3}} \right\}}{K^{1-\beta}} \right)
$$
$$
+ \delta K^3 \cdot \left( \frac{LR_0^2}{K} + \frac{\max\left\{ R_0 \sigma, L^{\frac{\alpha-1}{3\alpha-1}} R_0^{\frac{4\alpha-2}{3\alpha-1}} \sigma^{\frac{2\alpha}{3\alpha-1}}, L^{\frac{1}{3}} R_0^{\frac{4}{3}} \sigma^{\frac{2}{3}} \right\}}{K^{1-\beta}} \right).
$$

Choosing $\delta$ in an appropriate way, we get the following result.

**Corollary 2.** Let the conditions of Theorem 1 hold and $\delta = \frac{1}{K^3}$. Then, we have

$$\mathbb{E}[f(x_K) - f^*]$$

$$= \mathcal{O}\left( \frac{LR_0^2 \log^2\left(6(K+1)^5\right)}{K} + \frac{\max\left\{ R_0\sigma, L^{\frac{\alpha-1}{3\alpha-1}} R_0^{\frac{4\alpha-2}{3\alpha-1}} \sigma^{\frac{2\alpha}{3\alpha-1}}, L^{\frac{1}{3}} R_0^{\frac{4}{3}} \sigma^{\frac{2}{3}} \right\} \log^2\left(6(K+1)^5\right)}{K^{1-\beta}} \right).$$

**Remark 2.** Clearly, we can substitute the best possible choice of $\beta$ to obtain the same convergence rate as in Corollary 1.

**Remark 3.** In-expectation convergence from Corollary 2 reproduces the result from Taylor & Bach (2019) for $\alpha = 2$ up to logarithmic factors.

**Remark 4.** Since we take $\delta = \frac{1}{K^3}$, the resulting choice of $\gamma_k$ and $\lambda_k$ in Corollary 2 is horizon-dependent. Achieving a similar in-expectation convergence rate for Clipped-SGD horizon-agnostic hyper-parameters is an interesting open question.

**Remark 5.** Interestingly, upper bounds analogous to (7), which constitute the key ingredient in deriving the in-expectation guarantee from the high-probability results, can be established for *any method with bounded updates*, such as normalized SGD and SignSGD, with or without momentum. This demonstrates that the technique developed in this section extends well beyond Clipped-SGD and applies to a broad class of algorithms with bounded steps.

## 5 EXPERIMENTS

In this section, we present the results of numerical simulations showing the practical advantages of the last iterate over the average. We consider the problem of minimizing a convex and smooth function $f\colon \mathbb{R}^d \to \mathbb{R}$ from noisy estimates $\widehat{\nabla} f(x)$ of its gradients. In all experiments, we run Clipped-SGD with the step-size and clipping level schedules suggested by the theory, optimizing the constants via a grid-search procedure. Finally we report the performance of both the average and the last iterate in terms of the 0.95-percentile and, for completeness, also the sample mean $\pm$ the standard deviation of the function values across 1000 repetitions.

**Corrupted gradients.** We set $\widehat{\nabla} f(x) = \nabla f(x) + N$, where $N$ is a random vector with components sampled i.i.d. from a Pareto distribution rescaled and reshaped so that it satisfies $\mathbb{E}[\widehat{\nabla} f(x)] = \nabla f(x)$, $\mathbb{E}[\|\widehat{\nabla} f(x) - \nabla f(x)\|^2] \leq 1$, and all moments of order greater than 2.001 are infinite, which closely matches our assumption with $\alpha = 2$. We set $d = 100$ and consider two cases: first, for a fixed unit vector $a$ we let $f(x) = \ln(1 + \exp(\langle x, a\rangle)) + (\lambda/2)\|x\|^2$ (where $\lambda = 10^{-6}$, is only introduced to make the problem well defined); second, we also consider $f(x) = (1/2)\|x\|^2$. Both objectives are smooth and strongly convex, although the first one is almost not strongly convex due to the very small value of $\lambda$. The results are shown in the left and central plots of Figure 1, where it is possible to see that the last iterate performs better than the average.

**Statistical learning.** We also consider the following statistical learning problem, in which we aim to minimize

$$f(x) = \mathbb{E}_{(Z,Y)\sim\mu}\left[\ln\left(1 + \exp(-Y\langle x, Z\rangle)\right)\right] + \frac{\lambda}{2}\|x\|^2$$

using only sampling access to $\mu = \mu_Z \cdot \mu_{Y|Z}$ and $\lambda = 10^{-2}$ (again introduced only to make the problem well-defined). We set $d = 10$ and take $\mu_Z$ to be an isotropic distribution with components sampled from a Student-$t$ distribution with 2.001 degrees of freedom, so that each component has variance $1/d$ but infinite moments of higher order. Moreover, we set $\mu_{Y|Z=z} = \text{Ber}(p(z))$ over $\{\pm 1\}$ with $p(z) = \text{sigmoid}(\langle w, z\rangle)$. We use $\widehat{\nabla} f(x) = \nabla_x \ln(1 + \exp(-Y\langle x, Z\rangle))$ for $(Z, Y) \sim \mu$ and note that $\mathbb{E}[\widehat{\nabla} f(x)] = \nabla f(x)$. We estimate $f(z)$ at a given point $z$ via the Median-of-Means[5]

---

[5]The use of Median-of-Means ensures an estimation error of the order $\sqrt{\frac{\log(1/\beta)}{m}}$ with confidence $1 - \beta$ over $m$ samples, which is exponentially better than the $\sqrt{\frac{1}{\beta \cdot m}}$ exhibited by the standard Monte-Carlo estimate (Lugosi & Mendelson, 2019).

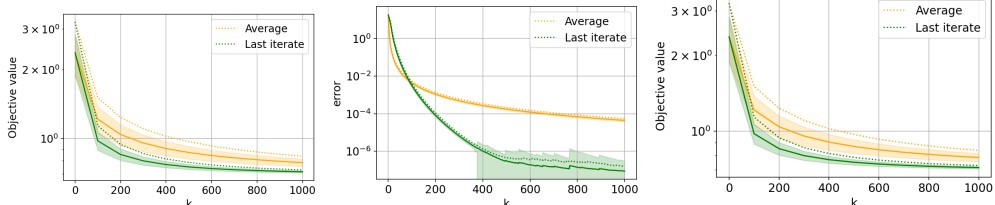

Figure 1: Experimental results: (left) $f(x) = \ln(1 + \exp(\langle x, a \rangle))$, (center) $f(x) = \frac{1}{2}\|x\|^2$, and (right) $f(x) = \mathbb{E}_{(Z,Y)\sim\mu}[\ln(1 + \exp(-Y\langle x, Z \rangle))]$. The $x$-axis shows the iteration counter $k$, while the $y$-axis (in logarithmic scale) reports the objective value $f(z_k)$ at the iterate $z_k$. Note that in the left and center plots, the $y$-axis corresponds to the optimization error since $\inf_{x\in\mathbb{R}^d} f(x) = 0$. In all figures, the dotted line denotes the $0.95$-percentile, while the solid line together with the shaded area denotes the sample mean $\pm$ one standard deviation. Note that, due to the log scale on the $y$, the shaded area appears stretched whenever its magnitude differs by orders of magnitude from the sample mean.

estimator with $10^3$ samples from $\mu$. The results are shown in the right plot of Figure 1, where it is possible to see that the last iterate performs better than the average.

**Discussion.** For fairness, both the average and the last iterate are evaluated under the same step-size and clipping level schedules: namely, those for which our theory guarantees convergence of the last iterate. We emphasize that the optimal schedules for the average iterate differ from these ones. In additional experiments (see appendix), we observed that the last iterate actually performs even better when the average iterate is run under its own optimal schedule. This suggests that our current $\widetilde{\mathcal{O}}(1/K^{1/3})$ bound for the last iterate may be an artifact of the analysis, and that there might exist a schedule (possibly the one already known to be optimal for the average iterate) that makes the last iterate achieve the optimal rate as well. A rigorous proof of this conjecture is left to future work.

## 6 CONCLUSION

We presented the first high-probability last-iterate guarantees for Clipped-SGD on convex $L$-smooth objectives under heavy-tailed noise with finite $\alpha$-th moments, $\alpha \in (1, 2]$. Our analysis is based on a potential function tailored to high-probability control, a new clipping schedule that scales as $1/(\sqrt{b_k}\gamma_k)$, and horizon-agnostic parameter choices. These ingredients yield a rate of $\widetilde{\mathcal{O}}(1/K^{2(\alpha-1)/3\alpha})$ for the last iterate with only polylogarithmic dependence on $1/\delta$. Empirically, we observe a clear advantage of the last iterate over the average under heavy-tailed perturbations.

**Limitations and future work.** The rate at $\alpha = 2$ leaves a polynomial gap from the $\widetilde{\mathcal{O}}(1/\sqrt{K})$ expectation benchmark (Liu & Zhou, 2024); tightening the last-iterate high-probability rate is a compelling direction. Extending the theory to $(L_0, L_1)$-smooth objectives, which is done for the *average iterate* by Gaash et al. (2025) under sub-Gaussian noise assumption and by Chezhegov et al. (2025) under Assumption 4, without losing horizon-freeness, is another natural next step.

## ACKNOWLEDGMENTS

The work of Savelii Chezhegov was supported by the Ministry of Economic Development of the Russian Federation (agreement No. 139-15-2025-013, dated June 20, 2025, IGK 000000C313925P4B0002).

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

# A APPENDIX

In this section, we provide additional known details intended to support our analysis.

## A.1 TECHNICAL DETAILS

**Additional notation.** We introduce the following notation:

$$
\begin{aligned}
g_t &= \mathrm{clip}\left(\nabla f_{\xi_t}(x_t), \lambda_t\right), \\
\theta_t &= g_t - \nabla f(x_t), \\
\theta_t^u &= g_t - \mathbb{E}_{\xi_t}[g_t], \\
\theta_t^b &= \mathbb{E}_{\xi_t}[g_t] - \nabla f(x_t), \\
R_t &= \|x_t - x^*\|, \\
\Delta_t &= f(x_t) - f^*.
\end{aligned}
$$

Moreover, we provide the main notation that appears in the analysis to simplify the reading of the proof:

- $\gamma_k$ – the stepsize at iteration $k$;
- $\lambda_k$ – the clipping threshold at iteration $k$;
- $d_k$ – the weight for the function distance between the current point $x_k$ and the optimal one, which allows to construct the potential $\Phi_k$;
- $\Phi_0$ – the initial potential, which is equal to $L\|x_0 - x^*\|^2$.

We also use the following standard results.

**Lemma 1** (Lemma 5.1 from (Sadiev et al., 2023)). *Let $X$ be a random vector from $\mathbb{R}^d$ and $\widehat{X} = \mathrm{clip}(X, \lambda)$. Then, $\left\|\widehat{X} - \mathbb{E}\left[\widehat{X}\right]\right\| \le 2\lambda$. Moreover, if for some $\sigma \ge 0$ and $\alpha \in (1, 2]$ we have $\mathbb{E}[X] = x \in \mathbb{R}^d$, $\mathbb{E}\left[\|X - x\|^\alpha\right] \le \sigma^\alpha$, and $\|x\| \le \frac{\lambda}{2}$, then*

$$
\left\|\mathbb{E}\left[\widehat{X}\right] - x\right\| \le \frac{2^\alpha \sigma^\alpha}{\lambda^{\alpha-1}},
$$

$$
\mathbb{E}\left[\left\|\widehat{X} - x\right\|^2\right] \le 18\lambda^{2-\alpha}\sigma^\alpha,
$$

$$
\mathbb{E}\left[\left\|\widehat{X} - \mathbb{E}\left[\widehat{X}\right]\right\|^2\right] \le 18\lambda^{2-\alpha}\sigma^\alpha.
$$

This lemma provides sufficient bounds for quantities such as the bias and variance of the clipped stochastic gradient, which satisfies Assumption 4.

Next, we use one of the most popular concentration inequalities: Bernstein's inequality (Bennett, 1962; Dzhaparidze & Van Zanten, 2001; Freedman et al., 1975).

**Lemma 2** (Bernstein's inequality). *Let the sequence of random variables $\{X_i\}_{i\ge1}$ form a martingale difference sequence, i.e., $\mathbb{E}\left[X_i \mid X_{i-1}, \ldots, X_1\right] = 0$ for all $i \ge 1$. Assume that conditional variances $\sigma_i^2 = \mathbb{E}\left[X_i^2 \mid X_{i-1}, \ldots, X_1\right]$ exist and are bounded and also assume that there exists deterministic constant $c > 0$ such that $|X_i| \le c$ almost surely for all $i \ge 1$. Then for all $b > 0$, $G > 0$ and $n \ge 1$*

$$
\mathbb{P}\left\{\left|\sum_{i=1}^n X_i\right| > b \text{ and } \sum_{i=1}^n \sigma_i^2 \le G\right\} \le 2\exp\left(-\frac{b^2}{2G + \frac{2cb}{3}}\right).
$$

Additionally, we formulate Young's inequality.

**Proposition 1** (Young's inequality.). *For any $x, y \in \mathbb{R}^d$ and $p > 0$ the following inequality holds:*

$$
\|x + y\|^2 \le (1 + p)\|x\|^2 + \left(1 + \frac{1}{p}\right)\|y\|^2.
$$

In particular, for $p = 1$

$$\|x + y\|^2 \leq 2\|x\|^2 + 2\|y\|^2.$$

# B  MISSING PROOFS

This section is organized as follows. First, we introduce an auxiliary numerical lemma required for the main proof. Next, we state the descent lemma, which serves as the foundation for deriving the main result. Finally, we present the convergence rate theorem for Clipped-SGD based on the last iterate.

## B.1  AUXILIARY NUMERICAL LEMMA

**Lemma 3** (Numerical lemma). Suppose that $t \geq 0$, $t$ is integer, $\beta \in (0, 1)$, $m \geq 0$ and $n > 0$. Then, we have

$$\left(\frac{(t+1)^{(1-\beta)} - 1}{1 - \beta}\right)^n (t+1)^{-m} \leq \frac{\ln^n(t+1)}{(t+1)^{m-(1-\beta)n}}.$$

*Proof.* Let us consider

$$\left(\frac{(t+1)^{1-\beta} - 1}{1 - \beta}\right)^n (t+1)^{-m}.$$

For the case $t = 0$ it is obvious: $0 \leq 0$. For $t \geq 1$, it can be rewritten as

$$\left(\frac{(t+1)^{1-\beta} - 1}{1 - \beta}\right)^n (t+1)^{-m} = \left(\ln(t+1)\frac{e^{(1-\beta)\ln(t+1)} - 1}{(1 - \beta)\ln(t+1)}\right)^n (t+1)^{-m}$$

$$= \frac{\ln^n(t+1)}{(t+1)^m}\left(\frac{e^{(1-\beta)\ln(t+1)} - 1}{(1 - \beta)\ln(t+1)}\right)^n.$$

What is more, it is known that for all $x > 0$ we have

$$\frac{e^x - 1}{x} \leq e^x.$$

It is enough to apply Taylor series or compare the growth of both parts. Therefore, with $\beta < 1$ and $\ln(t + 1) > 0$ since $t \geq 1$, we have

$$\left(\frac{(t+1)^{1-\beta} - 1}{1 - \beta}\right)^n (t+1)^{-m} = \frac{\ln^n(t+1)}{(t+1)^m}\left(\frac{e^{(1-\beta)\ln(t+1)} - 1}{(1 - \beta)\ln(t+1)}\right)^n$$

$$\leq \frac{\ln^n(t+1)}{(t+1)^m}\left(e^{(1-\beta)\ln(t+1)}\right)^n$$

$$= \frac{\ln^n(t+1)}{(t+1)^m}(t+1)^{(1-\beta)n}$$

$$= \frac{\ln^n(t+1)}{(t+1)^{m-(1-\beta)n}}.$$

This finishes the proof. □

## B.2  DESCENT LEMMA

**Lemma 4** (Descent lemma). Suppose that Assumptions 1, 2 and 3 hold. Then, if for all $k \geq 0$ we have $\gamma_k \leq \frac{1}{2L}$, after $K$ iterations of Clipped-SGD, we have

$$\Phi_K \leq \Phi_0 - \sum_{k=0}^{K-1} 2\gamma_k L \langle x_k - x^*, \theta_k \rangle - \sum_{k=0}^{K-1} d_k \gamma_k \langle \nabla f(x_k), \theta_k \rangle + \sum_{k=0}^{K-1} (d_{k+1} L \gamma_k^2 + L\gamma_k^2)\|\theta_k\|^2,$$

where $d_{k+1} := d_k + 2\gamma_k L$, $d_0 := 0$ and $\Phi_k := d_k(f(x_k) - f^*) + L\|x_k - x^*\|^2$.

*Proof.* We start with the definition of $\Phi_k$:

$$\Phi_{k+1} - \Phi_k = d_{k+1}(f(x_{k+1}) - f^*) + L\|x_{k+1} - x^*\|^2 - d_k(f(x_k) - f^*) - L\|x_k - x^*\|^2$$

$$= d_{k+1}(f(x_{k+1}) - f(x_k)) + (d_{k+1} - d_k)(f(x_k) - f^*)$$

$$+ L(\|x_{k+1} - x^*\|^2 - \|x_k - x^*\|^2).$$

Decomposing the last term allows, we obtain

$$\Phi_{k+1} - \Phi_k = d_{k+1}(f(x_{k+1}) - f(x_k)) + (d_{k+1} - d_k)(f(x_k) - f^*)$$
$$+ L(2 \langle x_{k+1} - x_k, x_k - x^* \rangle + \|x_{k+1} - x_k\|^2).$$

Applying $L$-smoothness to the first term and using the convexity for the second one, we get

$$\Phi_{k+1} - \Phi_k \le d_{k+1} \left( \langle \nabla f(x_k), x_{k+1} - x_k \rangle + \frac{L}{2} \|x_{k+1} - x_k\|^2 \right)$$
$$+ (d_{k+1} - d_k) \langle \nabla f(x_k), x_k - x^* \rangle$$
$$+ L(2 \langle x_{k+1} - x_k, x_k - x^* \rangle + \|x_{k+1} - x_k\|^2). \tag{8}$$

Next, substitution of the update rule for $x_{k+1}$ into (8) gives the following inequality;

$$\Phi_{k+1} - \Phi_k \le d_{k+1} \left( -\gamma_k \langle \nabla f(x_k), g_k \rangle + \frac{L\gamma_k^2}{2} \|g_k\|^2 \right)$$
$$+ (d_{k+1} - d_k) \langle \nabla f(x_k), x_k - x^* \rangle + L(-2\gamma_k \langle g_k, x_k - x^* \rangle + \gamma_k^2 \|g_k\|^2).$$

Therefore, using the notation of $\theta_k := g_k - \nabla f(x_k)$, we derive

$$\Phi_{k+1} - \Phi_k \le d_{k+1} \left( -\gamma_k \langle \nabla f(x_k), g_k \rangle + \frac{L\gamma_k^2}{2} \|g_k\|^2 \right)$$
$$+ (d_{k+1} - d_k) \langle \nabla f(x_k), x_k - x^* \rangle + L(-2\gamma_k \langle g_k, x_k - x^* \rangle + \gamma_k^2 \|g_k\|^2)$$
$$\le d_{k+1} \left( -\gamma_k \langle \nabla f(x_k), \theta_k \rangle - \gamma_k \|\nabla f(x_k)\|^2 + L\gamma_k^2 (\|\nabla f(x_k)\|^2 + \|\theta_k\|^2) \right)$$
$$+ (d_{k+1} - d_k - 2\gamma_k L) \langle \nabla f(x_k), x_k - x^* \rangle - 2\gamma_k L \langle \theta_k, x_k - x^* \rangle$$
$$+ L\gamma_k^2 (\|\nabla f(x_k)\|^2 + 2 \langle \nabla f(x_k), \theta_k \rangle + \|\theta_k\|^2).$$

Combining terms, we have

$$\Phi_{k+1} - \Phi_k \le \left( -d_{k+1}(\gamma_k - L\gamma_k^2) + L\gamma_k^2 \right) \|\nabla f(x_k)\|^2 - 2\gamma_k L \langle \theta_k, x_k - x^* \rangle$$
$$- \gamma_k(d_{k+1} - 2L\gamma_k) \langle \nabla f(x_k), \theta_k \rangle + (d_{k+1} L\gamma_k^2 + L\gamma_k^2)\|\theta_k\|^2,$$

where we use $d_{k+1} - d_k - 2\gamma_k L = 0$. What is more, since $\gamma_k \le \frac{1}{2L}$, we have $\gamma_k - L\gamma_k^2 \ge \frac{\gamma_k}{2}$. Therefore, using the update of $d_{k+1}$, we have $\left( -d_{k+1}(\gamma_k - L\gamma_k^2) + L\gamma_k^2 \right) \le -d_k \gamma_k$, and, as a result,

$$\Phi_{k+1} - \Phi_k \le -\frac{d_k \gamma_k}{2} \|\nabla f(x_k)\|^2 - 2\gamma_k L \langle \theta_k, x_k - x^* \rangle - d_k \gamma_k \langle \nabla f(x_k), \theta_k \rangle$$
$$+ (d_{k+1} L\gamma_k^2 + L\gamma_k^2)\|\theta_k\|^2$$
$$\le -2\gamma_k L \langle \theta_k, x_k - x^* \rangle - d_k \gamma_k \langle \nabla f(x_k), \theta_k \rangle + (d_{k+1} L\gamma_k^2 + L\gamma_k^2)\|\theta_k\|^2.$$

Summing up, we conclude the proof. □

### B.3 PROOF OF THEOREM 1

**Theorem 2.** *Suppose that Assumptions 1, 2, 3 and 4 hold. Then, if we choose*

$$\gamma_k = \min \left\{ \frac{1}{320L \ln \left( \frac{6(k+1)^2}{\delta} \right)}, \frac{B}{80 \cdot 13^{1/\alpha} (k+1)^\beta \ln \left( \frac{6(k+1)^2}{\delta} \right)} \right\}, \tag{9}$$

$$\lambda_k = \frac{R_0 \min \left\{ \frac{1}{\sqrt{d_k}}, 1 \right\}}{80\gamma_k \ln^{1/2} \left( \frac{6(k+1)^2}{\delta} \right)}, \tag{10}$$

*where $B$ is chosen as*

$$B = \min\left\{ \frac{R_0}{\sigma}, \frac{R_0^{\frac{2\alpha}{3\alpha-1}}}{\sigma^{\frac{2\alpha}{3\alpha-1}} L^{\frac{\alpha-1}{3\alpha-1}}}, \frac{R_0^{\frac{2}{3}}}{\sigma^{\frac{2}{3}} L^{\frac{1}{3}}} \right\}, \tag{11}$$

*parameter $\beta$ satisfies the following boundary:*

$$\beta \geq \frac{2+\alpha}{3\alpha};$$

*then, after $K$ iterations of* Clipped-SGD, *we have that*

$$f(x_K) - f^*$$
$$= \mathcal{O}\left( \left( \frac{LR_0^2}{K} + \frac{\max\left\{ R_0\sigma, L^{\frac{\alpha-1}{3\alpha-1}} R_0^{\frac{4\alpha-2}{3\alpha-1}} \sigma^{\frac{2\alpha}{3\alpha-1}}, L^{\frac{1}{3}} R_0^{\frac{4}{3}} \sigma^{\frac{2}{3}} \right\}}{K^{1-\beta}} \right) \ln^2\left( \frac{6(K+1)^2}{\delta} \right) \right)$$

*hold with probability at least $1 - \delta \sum_{t=1}^{K} \frac{1}{t^2}$.*

**Remark 6.** During the proof, some constrains on $\beta$ appear. This parameter obviously has a strong impact on a final convergence bound. This is why, in the course of the proof, parameter $\beta$ must be chosen with greater care. These constraints are formulated as **Conditions** throughout the proof. In the end of the proof, we show that $\beta \geq \frac{2+\alpha}{3\alpha}$ meets these conditions.

*Proof.* The proof in constructed in the similar manner as in Sadiev et al. (2023). For each $k = 0, 1, \dots$ let us consider probabilistic events $E_k$: inequalities

$$-\sum_{l=0}^{t-1} 2\gamma_l L \langle x_l - x^*, \theta_l \rangle - \sum_{l=0}^{t-1} d_l \gamma_l \langle \nabla f(x_l), \theta_l \rangle + \sum_{l=0}^{t-1} (d_{l+1} L\gamma_l^2 + L\gamma_l^2)\|\theta_l\|^2 \leq \Phi_0 \ln\left( \frac{6(t+1)^2}{\delta} \right);$$

$$\Phi_t \leq 2\Phi_0 \ln\left( \frac{6(t+1)^2}{\delta} \right)$$

hold for all $t = 0, \dots, k$ simultaneously. We want to show via induction that $\mathbb{P}\{E_k\} \geq 1 - \delta \sum_{t=1}^{k} \frac{1}{t^2}$. For $k = 0$, it is obvious since the left-hand side of the first inequality is zero and the second inequality holds for $t = 0$ since the log-factor is larger than 1. Now, let us assume that $\mathbb{P}\{E_{T-1}\} \geq 1 - \delta \sum_{t=1}^{T-1} \frac{1}{t^2}$ for some $T \geq 1$. Then, applying Lemma 4, we have

$$\Phi_t \leq \Phi_0 - \sum_{l=0}^{t-1} \gamma_l L \langle x_l - x^*, \theta_l \rangle - \sum_{l=0}^{t-1} d_l \gamma_l \langle \nabla f(x_l), \theta_l \rangle + \sum_{l=0}^{t-1} (d_{l+1} L\gamma_l^2 + L\gamma_l^2)\|\theta_l\|^2 \tag{12}$$

for all $t = 0, \dots, T$. Moreover, for all $t = 0, \dots, T-1$ the event $E_{T-1}$ implies

$$\Phi_t \leq \Phi_0 - \sum_{l=0}^{t-1} \gamma_l L \langle x_l - x^*, \theta_l \rangle - \sum_{l=0}^{t-1} d_l \gamma_l \langle \nabla f(x_l), \theta_l \rangle + \sum_{l=0}^{t-1} (d_{l+1} L\gamma_l^2 + L\gamma_l^2)\|\theta_l\|^2$$
$$\leq 2\Phi_0 \ln\left( \frac{6(t+1)^2}{\delta} \right).$$

At the same time, the event $E_{T-1}$ implies

$$\|\nabla f(x_t)\| \leq \begin{cases} \sqrt{2L(f(x_t) - f^*)}, & t > 0; \\ L\|x_t - x^*\|, & t \geq 0; \end{cases} \leq \begin{cases} \sqrt{\frac{4L^2 R_0^2 \ln\left( \frac{6(t+1)^2}{\delta} \right)}{d_t}}, & t > 0; \\ \sqrt{2L^2 R_0^2 \ln\left( \frac{6(t+1)^2}{\delta} \right)}, & t \geq 0; \end{cases},$$

where we use $d_t(f(x_t) - f^*) \leq \Phi_t \leq 2LR_0^2 \ln\left(\frac{6(t+1)^2}{\delta}\right)$. Moreover, the bound

$$\|\nabla f(x_0)\| \leq \frac{2LR_0\sqrt{\ln\left(\frac{6}{\delta}\right)}}{\sqrt{d_0}}$$

also holds since $d_0 = 0$ and we define $1/0 := +\infty$. Therefore, we get

$$\|\nabla f(x_t)\| \leq 2LR_0\sqrt{\ln\left(\frac{6(t+1)^2}{\delta}\right)} \min\left\{\frac{1}{\sqrt{d_k}}, 1\right\} \overset{(9),(10)}{\leq} \frac{\lambda_t}{2}. \tag{13}$$

Simultaneously, we have that $E_{T-1}$ implies

$$\|x_t - x^*\| \leq \sqrt{\frac{\Phi_t}{L}} \leq \sqrt{2}R_0\sqrt{\ln\left(\frac{6(t+1)^2}{\delta}\right)}. \tag{14}$$

Therefore, if we define random vectors $\eta_t$ and $\mu_t$ as

$$\eta_t = \begin{cases} \nabla f(x_t), & \|\nabla f(x_t)\| \leq 2LR_0\sqrt{\ln\left(\frac{6(t+1)^2}{\delta}\right)} \min\left\{\frac{1}{\sqrt{d_k}}, 1\right\}; \\ 0, & \text{otherwise,} \end{cases}$$

and

$$\mu_t = \begin{cases} x_t - x^*, & \|x_t - x^*\| \leq \sqrt{2}R_0\sqrt{\ln\left(\frac{6(t+1)^2}{\delta}\right)}; \\ 0, & \text{otherwise,} \end{cases}$$

then $\eta_t$ and $\mu_t$ can be upper bounded with probability 1. What is more, using (13) and (14), we can rewrite (12) using the new notation for truncated vectors and decomposing $\theta_t$ into bias and unbiased terms: $E_{T-1}$ implies

$$\Phi_T \leq \Phi_0 - \sum_{t=0}^{T-1} 2\gamma_t L \langle \mu_t, \theta_t \rangle - \sum_{t=0}^{T-1} d_t \gamma_t \langle \eta_t, \theta_t \rangle + \sum_{t=0}^{T-1} (d_{t+1}L\gamma_t^2 + L\gamma_t^2)\|\theta_t\|^2$$

$$\leq \Phi_0 - \underbrace{\sum_{t=0}^{T-1} 2\gamma_t L \langle \mu_t, \theta_t^u \rangle}_{①} - \underbrace{\sum_{t=0}^{T-1} 2\gamma_t L \langle \mu_t, \theta_t^b \rangle}_{②} - \underbrace{\sum_{t=0}^{T-1} d_t \gamma_t \langle \eta_t, \theta_t^u \rangle}_{③} - \underbrace{\sum_{t=0}^{T-1} d_t \gamma_t \langle \eta_t, \theta_t^b \rangle}_{④}$$

$$+ \underbrace{\sum_{t=0}^{T-1} 2(d_{t+1}L\gamma_t^2 + L\gamma_t^2)\left(\|\theta_t^u\|^2 - \mathbb{E}_{\xi_t}\left[\|\theta_t^u\|^2\right]\right)}_{⑤} + \underbrace{\sum_{t=0}^{T-1} 2(d_{t+1}L\gamma_t^2 + L\gamma_t^2)\mathbb{E}_{\xi_t}\left[\|\theta_t^u\|^2\right]}_{⑥}$$

$$+ \underbrace{\sum_{t=0}^{T-1} 2(d_{t+1}L\gamma_t^2 + L\gamma_t^2)\|\theta_t^b\|^2}_{⑦},$$

where we apply Young's inequality for $\|\theta_t^u + \theta_t^b\|^2$.

Next, we are going to derive a sufficient bound on each term: $①, \ldots, ⑦$. However, before we start, let us provide additional upper bounds for the sequence $\{d_t\}$. For the case $t = 0$ we have $d_t = 0$. On the other hand, for $t > 0$ we get

$$d_t = \sum_{k=0}^{t-1} 2\gamma_k L \overset{(9)}{\leq} \sum_{k=0}^{t-1} \frac{2BL}{80 \cdot 13^{1/\alpha}(k+1)^\beta} \leq \frac{2BL}{80 \cdot 13^{1/\alpha}}\left(1 + \int_1^t \frac{1}{x^\beta}dx\right)$$

$$\leq \frac{2BL}{80 \cdot 13^{1/\alpha}}\left(1 + \frac{t^{1-\beta} - 1}{1 - \beta}\right). \tag{15}$$

To reflect the correct behavior of convergence with $\alpha \to 1$, we need to bound $d_t$ as above. Nevertheless, to obtain the correct bound for fixed $\alpha \in (1, 2]$, it is *enough* to use

$$\frac{2BL}{80 \cdot 13^{1/\alpha}} \frac{t^{1-\beta}}{1-\beta}. \tag{16}$$

The problem arises when we consider the limit $\alpha \to 1$. In the next part of the proof it will be shown that

$$\alpha \to 1 \Rightarrow \beta \to 1$$

due to the constraints over $\beta$. Consequently, we could get

$$\lim_{\beta \to 1} \frac{t^{1-\beta}}{1-\beta} = \infty$$

instead of

$$\lim_{\beta \to 1} \frac{t^{1-\beta} - 1}{1-\beta} = \ln(t).$$

As a result, for more reasonable analysis, we *must* use (15) instead of (16).

Next, since the full gradients are upper-bounded by $\lambda_t$ (13), we can apply Lemma 1 to obtain that $E_{T-1}$ implies

$$\|\theta_t^u\| \le 2\lambda_t,$$

$$\|\theta_t^b\| \le \frac{2^\alpha \sigma^\alpha}{\lambda_t^{\alpha-1}},$$

$$\mathbb{E}_{\xi_t}\left[\|\theta_t^u\|^2\right] \le 18\lambda_t^{2-\alpha}\sigma^\alpha.$$

Now we are ready to bound terms ① − ⑦.

**Bound for** ①. By definition $\theta_t^u$, we get

$$\mathbb{E}_{\xi_t}[-2\gamma_t L \langle \mu_t, \theta_t^u \rangle] = 0.$$

Moreover, we have that

$$|-2\gamma_t L \langle \mu_t, \theta_t^u \rangle| \le 2\gamma_t L \|\mu_t\| \|\theta_t^u\| \le 6\gamma_t \lambda_t L R_0 \sqrt{\ln\left(\frac{6(t+1)^2}{\delta}\right)} \overset{(10)}{\le} \frac{6LR_0^2}{80} \le \frac{\Phi_0}{2} := c.$$

Next, we define $\sigma_t^2$:

$$\sigma_t^2 = \mathbb{E}_{\xi_t}[4\gamma_t^2 L^2 \langle \mu_t, \theta_t^u \rangle^2] \le 8\gamma_t^2 L^2 R_0^2 \ln\left(\frac{6(t+1)^2}{\delta}\right) \mathbb{E}_{\xi_t}\left[\|\theta_t^u\|^2\right].$$

Therefore, we can apply Bernstein's inequality with $b = \Phi_0 \ln\left(\frac{6(T)^2}{\delta}\right)/2$ and $G = \Phi_0^2 \ln\left(\frac{6(T)^2}{\delta}\right)/24$:

$$\mathbb{P}\left\{|①| > b \text{ and } \sum_{t=0}^{T-1} \sigma_t^2 \le G\right\} \le 2\exp\left(-\frac{b^2}{2G + \frac{2cb}{3}}\right) = \frac{\delta}{3T^2}.$$

Consequently, we have

$$\mathbb{P}\left\{|①| \le b \text{ either } \sum_{t=0}^{T-1} \sigma_t^2 > G\right\} \ge 1 - \frac{\delta}{3T^2}.$$

What is more, we have that $E_{T-1}$ implies

$$\sum_{t=0}^{T-1} \sigma_t^2 \leq \sum_{t=0}^{T-1} 8\gamma_t^2 L^2 R_0^2 \ln\left(\frac{6(t+1)^2}{\delta}\right) \mathbb{E}_{\xi_t}\left[\|\theta_t^u\|^2\right] \leq \sum_{t=0}^{T-1} 144\gamma_t^2 L^2 R_0^2 \ln\left(\frac{6(t+1)^2}{\delta}\right) \lambda_t^{2-\alpha} \sigma^\alpha$$

$$\overset{(10)}{\leq} \sum_{t=0}^{T-1} \frac{144 L^2 R_0^{4-\alpha} \gamma_t^\alpha \sigma^\alpha \ln^{\alpha/2}\left(\frac{6(t+1)^2}{\delta}\right)}{80^{2-\alpha}}.$$

Using the upper bound on $B$: $B \leq \frac{R_0}{\sigma}$ in the choice of $\gamma_t$ (9), we obtain that $E_{T-1}$ implies

$$\sum_{t=0}^{T-1} \sigma_t^2 \leq \sum_{t=0}^{T-1} \frac{144 L^2 R_0^{4-\alpha} \gamma_t^\alpha \sigma^\alpha \ln^{\alpha/2}\left(\frac{6(t+1)^2}{\delta}\right)}{80^{2-\alpha}} \leq \sum_{t=0}^{T-1} \frac{144 L^2 R_0^4}{13 \cdot 80^2 (t+1)^{\beta\alpha}}.$$

To obtain a sufficient bound depending on $\ln\left(\frac{6T^2}{\delta}\right)$, we must have this term in the power of *one*. Thus, we obtain

> **Condition 1:**
> $$\beta\alpha \geq 1 \Rightarrow \beta \geq \frac{1}{\alpha}. \tag{17}$$

Moreover, we get

$$\sum_{t=0}^{T-1} \frac{1}{(t+1)^{\beta\alpha}} \leq \sum_{t=0}^{T-1} \frac{1}{(t+1)} \leq 1 + \int_1^T \frac{1}{x}dx = 1 + \ln(T)$$

$$\leq 3\ln\left(\frac{\sqrt{6}T}{\sqrt{\delta}}\right) = 3/2 \ln\left(\frac{6T^2}{\delta}\right) \tag{18}$$

since $T \geq 1$. Therefore, with (17) we have that $E_{T-1}$ implies

$$\sum_{t=0}^{T-1} \sigma_t^2 \leq \sum_{t=0}^{T-1} \frac{144 L^2 R_0^4}{13 \cdot 80^2 (t+1)^{\beta\alpha}} \overset{(17),(18)}{\leq} \frac{\Phi_0^2 \ln\left(\frac{6T^2}{\delta}\right)}{24},$$

where we used that $LR_0^2 = \Phi_0$.

**Bound for ②.** The event $E_{T-1}$ implies

$$-\sum_{t=0}^{T-1} \gamma_t L \langle \mu_t, \theta_t^b \rangle \leq \sum_{t=0}^{T-1} \gamma_t L \|\mu_t\| \|\theta_t^b\| \leq \sum_{t=0}^{T-1} 2LR_0 \sqrt{\ln\left(\frac{6(t+1)^2}{\delta}\right)} \frac{2^\alpha \sigma^\alpha \gamma_t}{\lambda_t^{\alpha-1}}$$

$$= \sum_{t=0}^{T-1} 2LR_0^{2-\alpha} \ln^{\alpha/2}\left(\frac{6(t+1)^2}{\delta}\right) \cdot 2^\alpha 80^{\alpha-1} \sigma^\alpha \gamma_t^\alpha \max\{d_t, 1\}^{\frac{\alpha-1}{2}}$$

$$\leq \sum_{t=1}^{T-1} 2LR_0^{2-\alpha} \ln^{\alpha/2}\left(\frac{6(t+1)^2}{\delta}\right) \cdot 2^\alpha 80^{\alpha-1} \sigma^\alpha \gamma_t^\alpha d_t^{\frac{\alpha-1}{2}}$$

$$+ \sum_{t=0}^{T-1} 2LR_0^{2-\alpha} \ln^{\alpha/2}\left(\frac{6(t+1)^2}{\delta}\right) \cdot 2^\alpha 80^{\alpha-1} \sigma^\alpha \gamma_t^\alpha, \tag{19}$$

where we use $\max\{a,b\}^p \le a^p + b^p$ for $p \le 1$ and $d_0 = 0$. To handle the first sum, we apply (15) to obtain

$$\sum_{t=1}^{T-1} 2LR_0^{2-\alpha} \ln^{\alpha/2}\left(\frac{6(t+1)^2}{\delta}\right) \cdot 2^\alpha 80^{\alpha-1} \sigma^\alpha \gamma_t^\alpha d_t^{\frac{\alpha-1}{2}}$$

$$\overset{d_t <= d_{t+1}}{\le} \sum_{t=1}^{T-1} 2LR_0^{2-\alpha} \ln^{\alpha/2}\left(\frac{6(t+1)^2}{\delta}\right) \cdot 2^\alpha 80^{\alpha-1} \sigma^\alpha \gamma_t^\alpha d_{t+1}^{\frac{\alpha-1}{2}}$$

$$\le \sum_{t=1}^{T-1} 2LR_0^{2-\alpha} \ln^{\alpha/2}\left(\frac{6(t+1)^2}{\delta}\right) \cdot 2^\alpha 80^{\alpha-1} \sigma^\alpha \gamma_t^\alpha \cdot \left(\frac{2BL}{80\cdot 13^{1/\alpha}}\left(1 + \frac{(t+1)^{1-\beta}-1}{1-\beta}\right)\right)^{\frac{\alpha-1}{2}}$$

$$\le \sum_{t=1}^{T-1} 2LR_0^{2-\alpha} \ln^{\alpha/2}\left(\frac{6(t+1)^2}{\delta}\right) \cdot 2^\alpha 80^{\alpha-1} \sigma^\alpha \gamma_t^\alpha \cdot \left(\frac{6BL}{80\cdot 13^{1/\alpha}}\frac{(t+1)^{1-\beta}-1}{1-\beta}\right)^{\frac{\alpha-1}{2}},$$

where in the last inequality we use

$$1 + \frac{t^{1-\beta}-1}{1-\beta} \le 1 + \frac{(t+1)^{1-\beta}-1}{1-\beta} \le 3\frac{(t+1)^{1-\beta}-1}{1-\beta} \tag{20}$$

with $t \ge 1$. Thus, using (9) and $B \le \dfrac{R_0^{\frac{2\alpha}{3\alpha-1}}}{\sigma^{\frac{2\alpha}{3\alpha-1}} L^{\frac{\alpha-1}{3\alpha-1}}}$, we get

$$\sum_{t=1}^{T-1} 2LR_0^{2-\alpha} \ln^{\alpha/2}\left(\frac{6(t+1)^2}{\delta}\right) \cdot 2^\alpha 80^{\alpha-1} \sigma^\alpha \gamma_t^\alpha d_t^{\frac{\alpha-1}{2}}$$

$$\le \sum_{t=1}^{T-1} \frac{6\cdot 2^\alpha 80^{\alpha-1} LR_0^2}{(80\cdot 13^{1/\alpha})^{\frac{3\alpha-1}{2}} \ln^{\alpha/2}\left(\frac{6(t+1)^2}{\delta}\right)} \cdot \left(\frac{(t+1)^{1-\beta}-1}{1-\beta}\right)^{\frac{\alpha-1}{2}} (t+1)^{-\beta\alpha}.$$

Consequently, applying Lemma 3, we derive

$$\sum_{t=1}^{T-1} 2LR_0^{2-\alpha} \ln^{\alpha/2}\left(\frac{6(t+1)^2}{\delta}\right) \cdot 2^\alpha 80^{\alpha-1} \sigma^\alpha \gamma_t^\alpha d_t^{\frac{\alpha-1}{2}}$$

$$\le \sum_{t=1}^{T-1} \frac{6\cdot 2^\alpha 80^{\alpha-1} LR_0^2}{(80\cdot 13^{1/\alpha})^{\frac{3\alpha-1}{2}} \ln^{\alpha/2}\left(\frac{6(t+1)^2}{\delta}\right)} \cdot \left(\frac{(t+1)^{1-\beta}-1}{1-\beta}\right)^{\frac{\alpha-1}{2}} (t+1)^{-\beta\alpha}$$

$$\le \sum_{t=0}^{T-1} \frac{6\cdot 2^\alpha 80^{\alpha-1} LR_0^2}{(80\cdot 13^{1/\alpha})^{\frac{3\alpha-1}{2}} (t+1)^{\beta\alpha-(1-\beta)(\alpha-1)/2}}.$$

where in the last inequality we expand the sum limits and use that $\ln^{\frac{\alpha-1}{2}}(t+1) \le \ln^{\frac{\alpha}{2}}\left(\frac{6(t+1)^2}{\delta}\right)$. For the second sum in the right-hand side of (19), we substitute $\gamma_t$ with $B \le \frac{R_0}{\sigma}$:

$$\sum_{t=0}^{T-1} 2LR_0^{2-\alpha} \ln^{\alpha/2}\left(\frac{6(t+1)^2}{\delta}\right) \cdot 2^\alpha 80^{\alpha-1} \sigma^\alpha \gamma_t^\alpha \le \sum_{t=0}^{T-1} \frac{2LR_0^2 \cdot 2^\alpha 80^{\alpha-1}}{(80\cdot 13^{1/\alpha})^\alpha (t+1)^{\beta\alpha}}.$$

Using the inequality on $\sum\limits_{t=0}^{T-1} \frac{1}{t+1}$ from the previous bound with next conditions:

**Condition 2:**

$$\begin{cases} \beta\alpha - \frac{(1-\beta)(\alpha-1)}{2} \ge 1 \Rightarrow \beta \ge \frac{\alpha+1}{3\alpha-1}; \\ \beta\alpha \ge 1 \Rightarrow \beta \ge \frac{1}{\alpha}, \end{cases} \tag{21}$$

we conclude that $E_{T-1}$ implies

$$② \leq \sum_{t=0}^{T-1} \left( \frac{6 \cdot 2^\alpha 80^{\alpha-1} LR_0^2}{(80 \cdot 13^{1/\alpha})^{\frac{3\alpha-1}{2}}(t+1)} + \frac{2LR_0^2 \cdot 2^\alpha 80^{\alpha-1}}{(80 \cdot 13^{1/\alpha})^\alpha(t+1)} \right) \leq \frac{\Phi_0 \ln\left(\frac{6T^2}{\delta}\right)}{12},$$

where $LR_0^2 = \Phi_0$ due to the notation.

**Bound for ③.** Due to the definition $\theta_t^u$, we get

$$\mathbb{E}_{\xi_t} \left[ -d_t\gamma_t \langle \eta_t, \theta_t^u \rangle \right] = 0.$$

Moreover, it is uniformly bounded:

$$|-d_t\gamma_t \langle \eta_t, \theta_t^u \rangle| \leq d_t\gamma_t \|\eta_t\| \|\theta_t^u\| \leq 4d_t\gamma_t\lambda_t LR_0 \sqrt{\frac{\ln\left(\frac{6(t+1)^2}{\delta}\right)}{d_t}} \stackrel{(10)}{\leq} \frac{4LR_0^2}{80} \leq \frac{3\Phi_0}{32} := c.$$

Moreover, there is *no contradiction* for the case $t = 0$: the corresponding term is equal to 0 due to $d_0 = 0$. Let us define $\sigma_t^2 = \mathbb{E}_{\xi_t} \left[ (d_t\gamma_t \langle \eta_t, \theta_t^u \rangle)^2 \right]$. Hence, we get

$$\sigma_t^2 \leq \mathbb{E}_{\xi_t} \left[ d_t^2\gamma_t^2 \|\eta_t\|^2 \|\theta_t^u\|^2 \right] = d_t^2\gamma_t^2 \|\eta_t\|^2 \mathbb{E}_{\xi_t} \left[ \|\theta_t^u\|^2 \right].$$

Consequently, we can apply Bernstein's inequality with $b = \Phi_0 \ln\left(\frac{6(T)^2}{\delta}\right)/12$ and $G = \Phi_0^2 \ln\left(\frac{6(T)^2}{\delta}\right)/1152$:

$$\mathbb{P}\left\{ |③| > b \text{ and } \sum_{t=0}^{T-1} \sigma_t^2 \leq G \right\} \leq 2\exp\left( -\frac{b^2}{2G + \frac{2cb}{3}} \right) = \frac{\delta}{3T^2}.$$

It automatically leads to

$$\mathbb{P}\left\{ |③| \leq b \text{ either } \sum_{t=0}^{T-1} \sigma_t^2 > G \right\} \geq 1 - \frac{\delta}{3T^2}.$$

What is more, we get that $E_{T-1}$ implies

$$\sum_{t=0}^{T-1} \sigma_t^2 \leq \sum_{t=0}^{T-1} d_t^2\gamma_t^2 \|\eta_t\|^2 \mathbb{E}_{\xi_t} \left[ \|\theta_t^u\|^2 \right] \leq \sum_{t=1}^{T-1} 72d_t\gamma_t^2 L^2 R_0^2 \lambda_t^{2-\alpha}\sigma^\alpha \ln\left( \frac{6(t+1)^2}{\delta} \right)$$

$$\stackrel{(10)}{\leq} \sum_{t=1}^{T-1} \frac{72d_{t+1}^{\alpha/2}\gamma_t^\alpha L^2 R_0^{4-\alpha}\sigma^\alpha \ln^{\alpha/2}\left( \frac{6(t+1)^2}{\delta} \right)}{80^{2-\alpha}},$$

where in the first line we apply $d_0 = 0$. Using the upper bound (15) on $d_t$ and (20), we obtain that $E_{T-1}$ implies

$$\sum_{t=0}^{T-1} \sigma_t^2 \leq \sum_{t=1}^{T-1} \frac{72\gamma_t^\alpha L^2 R_0^{4-\alpha}\sigma^\alpha \ln^{\alpha/2}\left( \frac{6(t+1)^2}{\delta} \right)}{80^{2-\alpha}} \cdot \left( \frac{6BL}{80 \cdot 13^{1/\alpha}} \frac{(t+1)^{1-\beta} - 1}{1 - \beta} \right)^{\frac{\alpha}{2}}.$$

Therefore, substituting $\gamma_t$ with $B \leq \frac{R_0^{\frac{2}{3}}}{\sigma^{\frac{2}{3}}L^{\frac{1}{3}}}$ allows to get that $E_{T-1}$ implies

$$\sum_{t=0}^{T-1} \sigma_t^2 \leq \sum_{t=1}^{T-1} \frac{72\gamma_t^\alpha L^2 R_0^{4-\alpha}\sigma^\alpha \ln^{\alpha/2}\left( \frac{6(t+1)^2}{\delta} \right)}{80^{2-\alpha}} \cdot \left( \frac{6BL}{80 \cdot 13^{1/\alpha}} \frac{(t+1)^{1-\beta} - 1}{1 - \beta} \right)^{\frac{\alpha}{2}}$$

$$\leq \sum_{t=1}^{T-1} \frac{432L^2 R_0^4}{80^{2-\alpha}(80 \cdot 13^{1/\alpha})^{\frac{3\alpha}{2}} \ln^{\alpha/2}\left( \frac{6(t+1)^2}{\delta} \right)} \cdot \left( \frac{(t+1)^{1-\beta} - 1}{1 - \beta} \right)^{\frac{\alpha}{2}} (t+1)^{-\beta\alpha}.$$

Applying Lemma 3, we obtain that $E_{T-1}$ implies

$$\sum_{t=0}^{T-1} \sigma_t^2 \le \sum_{t=1}^{T-1} \frac{432 L^2 R_0^4}{80^{2-\alpha}(80 \cdot 13^{1/\alpha})^{\frac{3\alpha}{2}}(t+1)^{\beta\alpha-(1-\beta)\alpha/2}}.$$

Thus, conditions over $\beta$ can be formulated as follows:

**Condition 3:**

$$\beta\alpha - (1-\beta)\alpha/2 \ge 1 \Rightarrow \beta \ge \frac{2+\alpha}{3\alpha}. \tag{22}$$

Consequently, with (22), we have

$$\sum_{t=0}^{T-1} \sigma_t^2 \overset{(22)}{\le} \sum_{t=0}^{T-1} \frac{432 L^2 R_0^4}{80^{2-\alpha}(80 \cdot 13^{1/\alpha})^{\frac{3\alpha}{2}}(t+1)} \overset{(18)}{\le} \frac{648 \Phi_0^2 \ln\left(\frac{6T^2}{\delta}\right)}{80^{2-\alpha}(80 \cdot 13^{1/\alpha})^{\frac{3\alpha}{2}}} \le \frac{\Phi_0^2 \ln\left(\frac{6T^2}{\delta}\right)}{1152}.$$

**Bound for ④.** The event $E_{T-1}$ implies

$$-\sum_{t=0}^{T-1} d_t \gamma_t \left\langle \eta_t, \theta_t^b \right\rangle \le \sum_{t=1}^{T-1} d_t \gamma_t \|\eta_t\| \|\theta_t^b\|$$

$$\le \sum_{t=1}^{T-1} 2 d_t \gamma_t L R_0 \sqrt{\frac{\ln\left(\frac{6(t+1)^2}{\delta}\right)}{d_t}} \frac{2^\alpha \sigma^\alpha}{\lambda_t^{\alpha-1}}$$

$$\overset{(10)}{\le} \sum_{t=1}^{T-1} 2 \cdot 2^\alpha \cdot 80^{\alpha-1} d_t^{1/2} \gamma_t^\alpha L R_0^{2-\alpha} \sigma^\alpha \ln^{\alpha/2}\left(\frac{6(t+1)^2}{\delta}\right) \max\{d_t, 1\}^{\frac{\alpha-1}{2}}$$

$$\le \sum_{t=1}^{T-1} 2 \cdot 2^\alpha \cdot 80^{\alpha-1} d_t^{\alpha/2} \gamma_t^\alpha L R_0^{2-\alpha} \sigma^\alpha \ln^{\alpha/2}\left(\frac{6(t+1)^2}{\delta}\right)$$

$$+ \sum_{t=1}^{T-1} 2 \cdot 2^\alpha \cdot 80^{\alpha-1} \gamma_t^\alpha L R_0^{2-\alpha} \sigma^\alpha \ln^{\alpha/2}\left(\frac{6(t+1)^2}{\delta}\right),$$

where we use $d_0 = 0$ and $d_t^{1/2} \max\{d_t, 1\}^{\frac{\alpha-1}{2}} \le \max\{d_t, 1\}^{\frac{\alpha}{2}}$: if $d_t \le 1$, we have $d_t^{1/2} \max\{d_t, 1\}^{\frac{\alpha-1}{2}} \le 1$; otherwise we have $d_t^{1/2} \max\{d_t, 1\}^{\frac{\alpha-1}{2}} \le d_t^{\alpha/2}$. Hence, to bound the first term, we apply $d_t \le d_{t+1}$ and substitute the upper bound on $d_t$ from (15):

$$④ \le \sum_{t=1}^{T-1} 2 \cdot 2^\alpha \cdot 80^{\alpha-1} d_{t+1}^{\alpha/2} \gamma_t^\alpha L R_0^{2-\alpha} \sigma^\alpha \ln^{\alpha/2}\left(\frac{6(t+1)^2}{\delta}\right)$$

$$+ \sum_{t=1}^{T-1} 2 \cdot 2^\alpha \cdot 80^{\alpha-1} \gamma_t^\alpha L R_0^{2-\alpha} \sigma^\alpha \ln^{\alpha/2}\left(\frac{6(t+1)^2}{\delta}\right)$$

$$\le \sum_{t=1}^{T-1} 2 \cdot 2^\alpha \cdot 80^{\alpha-1} \gamma_t^\alpha L R_0^{2-\alpha} \sigma^\alpha \ln^{\alpha/2}\left(\frac{6(t+1)^2}{\delta}\right) \cdot \left(\frac{6BL}{80 \cdot 13^{1/\alpha}} \frac{(t+1)^{1-\beta}-1}{1-\beta}\right)^{\frac{\alpha}{2}}$$

$$+ \sum_{t=1}^{T-1} 2 \cdot 2^\alpha \cdot 80^{\alpha-1} \gamma_t^\alpha L R_0^{2-\alpha} \sigma^\alpha \ln^{\alpha/2}\left(\frac{6(t+1)^2}{\delta}\right),$$

where we bound $1 + \frac{(t+1)^{1-\beta}-1}{1-\beta}$ in the similar manner as for previous terms with $t \ge 1$. Now, we can substitute the choice of $\gamma_t$ from (9) with the upper bound on $B$: $B \le \frac{R_0^{\frac{2}{3}}}{\sigma^{\frac{2}{3}} L^{\frac{1}{3}}}$ for the first term,

and $B \leq \frac{R_0}{\sigma}$ for the second term, to derive

$$
\begin{aligned}
④ &\leq \sum_{t=1}^{T-1} 2 \cdot 2^\alpha \cdot 80^{\alpha-1} \gamma_t^\alpha L R_0^{2-\alpha} \sigma^\alpha \ln^{\alpha/2} \left( \frac{6(t+1)^2}{\delta} \right) \cdot \left( \frac{6BL}{80 \cdot 13^{1/\alpha}} \frac{(t+1)^{1-\beta}-1}{1-\beta} \right)^{\frac{\alpha}{2}} \\
&+ \sum_{t=1}^{T-1} 2 \cdot 2^\alpha \cdot 80^{\alpha-1} \gamma_t^\alpha L R_0^{2-\alpha} \sigma^\alpha \ln^{\alpha/2} \left( \frac{6(t+1)^2}{\delta} \right) \\
&\leq \sum_{t=1}^{T-1} \frac{12 \cdot 2^\alpha \cdot 80^{\alpha-1} L R_0^2}{(80 \cdot 13^{1/\alpha})^{\frac{3\alpha}{2}} \ln^{\alpha/2} \left( \frac{6(t+1)^2}{\delta} \right)} \cdot \left( \frac{(t+1)^{1-\beta}-1}{1-\beta} \right)^{\frac{\alpha}{2}} (t+1)^{-\beta\alpha} \\
&+ \sum_{t=1}^{T-1} \frac{2 \cdot 2^\alpha \cdot 80^{\alpha-1} L R_0^2}{(80 \cdot 13^{1/\alpha})^\alpha} (t+1)^{-\beta\alpha}.
\end{aligned}
$$

Applying Lemma 3, we get

$$
\begin{aligned}
④ &\leq \sum_{t=1}^{T-1} \frac{12 \cdot 2^\alpha \cdot 80^{\alpha-1} L R_0^2}{(80 \cdot 13^{1/\alpha})^{\frac{3\alpha}{2}} \ln^{\alpha/2} \left( \frac{6(t+1)^2}{\delta} \right)} \cdot \left( \frac{(t+1)^{1-\beta}-1}{1-\beta} \right)^{\frac{\alpha}{2}} (t+1)^{-\beta\alpha} \\
&+ \sum_{t=1}^{T-1} \frac{2 \cdot 2^\alpha \cdot 80^{\alpha-1} L R_0^2}{(80 \cdot 13^{1/\alpha})^\alpha} (t+1)^{-\beta\alpha} \\
&\leq \sum_{t=1}^{T-1} \frac{12 \cdot 2^\alpha \cdot 80^{\alpha-1} L R_0^2}{(80 \cdot 13^{1/\alpha})^{\frac{3\alpha}{2}} (t+1)^{\beta\alpha-(1-\beta)\alpha/2}} + \sum_{t=1}^{T-1} \frac{2 \cdot 2^\alpha \cdot 80^{\alpha-1} L R_0^2}{(80 \cdot 13^{1/\alpha})^\alpha (t+1)^{\beta\alpha}}.
\end{aligned}
$$

To derive a sufficient boundary, it is enough to apply next conditions over $\beta$:

**Condition 4:**

$$
\begin{cases}
\beta\alpha \geq 1 \Rightarrow \beta \geq \frac{1}{\alpha} \\
\beta\alpha - (1-\beta)\alpha/2 \geq 1 \Rightarrow \beta \geq \frac{2+\alpha}{3\alpha}.
\end{cases}
\tag{23}
$$

Hence, we have

$$
④ \overset{(23)}{\leq} \sum_{t=0}^{T-1} \frac{12 \cdot 2^\alpha \cdot 80^{\alpha-1} L R_0^2}{(80 \cdot 13^{1/\alpha})^{\frac{3\alpha}{2}} (t+1)} + \sum_{t=1}^{T-1} \frac{2 \cdot 2^\alpha \cdot 80^{\alpha-1} L R_0^2}{(80 \cdot 13^{1/\alpha})^\alpha (t+1)} \overset{(18)}{\leq} \frac{\Phi_0 \ln \left( \frac{6T^2}{\delta} \right)}{12},
$$

where we expand the sum limits and abuse the notation of $\Phi_0$.

**Bound for ⑤.** At first, we have

$$
\mathbb{E}_{\xi_t} \left[ 2(d_{t+1} L \gamma_t^2 + L \gamma_t^2) \left( \|\theta_t^u\|^2 - \mathbb{E}_{\xi_t} \left[ \|\theta_t^u\|^2 \right] \right) \right] = 0
$$

Moreover, we get that

$$
\left| 2(d_{t+1} L \gamma_t^2 + L \gamma_t^2) \left( \|\theta_t^u\|^2 - \mathbb{E}_{\xi_t} \left[ \|\theta_t^u\|^2 \right] \right) \right| \leq 8(d_{t+1} L \gamma_t^2 + L \gamma_t^2) \lambda_t^2 \overset{L\gamma_t \leq 1}{\leq} (8 d_t L \gamma_t^2 + 24 L \gamma_t^2) \lambda_t^2
$$

$$
\overset{(10)}{\leq} \frac{32 \Phi_0}{6400} \leq \frac{\Phi_0}{8} := c.
$$

Let us define $\sigma_t^2 = \mathbb{E}_{\xi_t} \left[ 4 e_t^2 \left( \|\theta_t^u\|^2 - \mathbb{E}_{\xi_t} \left[ \|\theta_t^u\|^2 \right] \right)^2 \right]$. Consequently, it can be bounded as follows:

$$
\begin{aligned}
\sigma_t^2 &= \mathbb{E}_{\xi_t} \left[ 4(d_{t+1} L \gamma_t^2 + L \gamma_t^2)^2 \left( \|\theta_t^u\|^2 - \mathbb{E}_{\xi_t} \left[ \|\theta_t^u\|^2 \right] \right)^2 \right] \\
&\leq c \mathbb{E}_{\xi_t} \left[ 2(d_{t+1} L \gamma_t^2 + L \gamma_t^2) \left| \left( \|\theta_t^u\|^2 - \mathbb{E}_{\xi_t} \left[ \|\theta_t^u\|^2 \right] \right) \right| \right] \\
&\leq 4c(d_{t+1} L \gamma_t^2 + L \gamma_t^2) \mathbb{E}_{\xi_t} \left[ \|\theta_t^u\|^2 \right].
\end{aligned}
$$

Therefore, we can apply Bernstein's inequality with $b = \Phi_0 \ln\left(\frac{6T^2}{\delta}\right)/8$ and $G = \Phi_0^2 \ln\left(\frac{6T^2}{\delta}\right)/384$:

$$\mathbb{P}\left\{|⑤| > b \text{ and } \sum_{t=0}^{T-1} \sigma_t^2 \leq G\right\} \leq 2\exp\left(-\frac{b^2}{2G + \frac{2cb}{3}}\right) = \frac{\delta}{3T^2}.$$

Hence, we derive

$$\mathbb{P}\left\{|⑤| \leq b \text{ either } \sum_{t=0}^{T-1} \sigma_t^2 > G\right\} \geq 1 - \frac{\delta}{3T^2}.$$

What is more, we get

$$\sum_{t=0}^{T-1} \sigma_t^2 \leq \sum_{t=0}^{T-1} 4c(d_{t+1}L\gamma_t^2 + L\gamma_t^2)\mathbb{E}_{\xi_t}\left[\|\theta_t^u\|^2\right] \overset{d_{t+1}=d_t+2L\gamma_t}{\leq} \sum_{t=1}^{T-1} 72c(d_t L\gamma_t^2 + 3L\gamma_t^2)\lambda_t^{2-\alpha}\sigma^\alpha$$

$$= \sum_{t=0}^{T-1} 72cd_t L\gamma_t^2 \lambda_t^{2-\alpha}\sigma^\alpha + \sum_{t=0}^{T-1} 216cL\gamma_t^2 \lambda_t^{2-\alpha}\sigma^\alpha, \tag{24}$$

where in the last equation we use $d_0 = 0$. To deal with the first term of (24), we substitute (10) to obtain

$$\sum_{t=1}^{T-1} 72cd_t L\gamma_t^2 \lambda_t^{2-\alpha}\sigma^\alpha \overset{(10)}{\leq} \sum_{t=1}^{T-1} \frac{72cd_t^{\alpha/2} LR_0^{2-\alpha}\gamma_t^\alpha\sigma^\alpha}{80^{2-\alpha} \ln^{1-\alpha/2}\left(\frac{6(t+1)^2}{\delta}\right)} \leq \sum_{t=1}^{T-1} \frac{72cd_{t+1}^{\alpha/2} LR_0^{2-\alpha}\gamma_t^\alpha\sigma^\alpha}{80^{2-\alpha} \ln^{1-\alpha/2}\left(\frac{6(t+1)^2}{\delta}\right)}.$$

Next, we apply (15) to substitute the upper bound on $d_{t+1}$:

$$\sum_{t=1}^{T-1} 72cd_t L\gamma_t^2 \lambda_t^{2-\alpha}\sigma^\alpha \leq \sum_{t=1}^{T-1} \frac{72cd_{t+1}^{\alpha/2} LR_0^{2-\alpha}\gamma_t^\alpha\sigma^\alpha}{80^{2-\alpha} \ln^{1-\alpha/2}\left(\frac{6(t+1)^2}{\delta}\right)}$$

$$\leq \sum_{t=1}^{T-1} \frac{72cLR_0^{2-\alpha}\gamma_t^\alpha\sigma^\alpha}{80^{2-\alpha} \ln^{1-\alpha/2}\left(\frac{6(t+1)^2}{\delta}\right)} \cdot \left(\frac{6BL}{80\cdot 13^{1/\alpha}} \frac{(t+1)^{1-\beta}-1}{1-\beta}\right)^{\alpha/2},$$

where we also use the idea that $1 + \frac{(t+1)^{1-\beta}-1}{1-\beta} \leq 3\frac{(t+1)^{1-\beta}-1}{1-\beta}$ for all $t \geq 1$. Using the choice of $\gamma_t$ (9), one can derive

$$\sum_{t=1}^{T-1} 72cd_t L\gamma_t^2 \lambda_t^{2-\alpha}\sigma^\alpha \leq \sum_{t=1}^{T-1} \frac{72cLR_0^{2-\alpha}\gamma_t^\alpha\sigma^\alpha}{80^{2-\alpha} \ln^{1-\alpha/2}\left(\frac{6(t+1)^2}{\delta}\right)} \cdot \left(\frac{6BL}{80\cdot 13^{1/\alpha}} \frac{(t+1)^{1-\beta}-1}{1-\beta}\right)^{\alpha/2}$$

$$\leq \sum_{t=1}^{T-1} \frac{432cLR_0^2}{80^{2-\alpha}(80\cdot 13^{1/\alpha})^{\frac{3\alpha}{2}} \ln^{1+\alpha/2}\left(\frac{6(t+1)^2}{\delta}\right)}$$

$$\cdot \left(\frac{(t+1)^{1-\beta}-1}{1-\beta}\right)^{\alpha/2} (t+1)^{-\beta\alpha},$$

where in the last inequality we apply $B \leq \frac{R_0^{\frac{2}{3}}}{\sigma^{\frac{2}{3}} L^{\frac{1}{3}}}$. Thus, noting that the RHS can be bounded according to Lemma 3, we get

$$\sum_{t=1}^{T-1} 72cd_t L\gamma_t^2 \lambda_t^{2-\alpha}\sigma^\alpha \leq \sum_{t=1}^{T-1} \frac{432cLR_0^2}{80^{2-\alpha}(80\cdot 13^{1/\alpha})^{\frac{3\alpha}{2}} \ln^{1+\alpha/2}\left(\frac{6(t+1)^2}{\delta}\right)} \tag{25}$$

$$\cdot \left(\frac{(t+1)^{1-\beta}-1}{1-\beta}\right)^{\alpha/2} (t+1)^{-\beta\alpha}$$

$$\leq \sum_{t=1}^{T-1} \frac{432cLR_0^2}{80^{2-\alpha}(80\cdot 13^{1/\alpha})^{\frac{3\alpha}{2}} (t+1)^{\beta\alpha-(1-\beta)\alpha/2}}. \tag{26}$$

To handle with the second term from (24), we substitute the upper bound on $\lambda_t$ (10) and $\gamma_t$ with $B \leq \frac{R_0}{\sigma}$:

$$\sum_{t=0}^{T-1} 216cL\gamma_t^2\lambda_t^{2-\alpha}\sigma^\alpha \overset{(10)}{\leq} \sum_{t=0}^{T-1} \frac{216cL\gamma_t^\alpha R_0^{2-\alpha}\sigma^\alpha}{80^{2-\alpha}\ln^{1-\alpha/2}\left(\frac{6(t+1)^2}{\delta}\right)} \overset{(9)}{\leq} \sum_{t=0}^{T-1} \frac{216cLR_0^2}{80^2 \cdot 13(t+1)^{\beta\alpha}}. \tag{27}$$

Thus, if we choose $\beta$ as follows:

**Condition 5:**

$$\begin{cases} \beta\alpha \geq 1 \Rightarrow \beta \geq \frac{1}{\alpha}; \\ \beta\alpha - (1-\beta)\alpha/2 \geq 1 \Rightarrow \beta \geq \frac{2+\alpha}{3\alpha}, \end{cases} \tag{28}$$

combining (25), (27) and (28), we derive

$$\sum_{t=0}^{T-1}\sigma_t^2 \overset{(28)}{\leq} \sum_{t=1}^{T-1} \frac{432cLR_0^2}{80^{2-\alpha}(80\cdot13^{1/\alpha})^{\frac{3\alpha}{2}}(t+1)} + \sum_{t=0}^{T-1}\frac{216cLR_0^2}{80^2\cdot13(t+1)} \overset{(18)}{\leq} \frac{\Phi_0^2\ln\left(\frac{6T^2}{\delta}\right)}{384},$$

where we expand the sum limits for the first term, apply (18) to bound $\sum\frac{1}{t+1}$, substitute the value $c$ and abuse the notation of $\Phi_0$.

**Bound for ⑥.** The event $E_{T-1}$ implies

$$\sum_{t=0}^{T-1} 2(d_{t+1}L\gamma_t^2 + L\gamma_t^2)\mathbb{E}_{\xi_t}\left[\|\theta_t^u\|^2\right] \leq \sum_{t=0}^{T-1} 36(d_{t+1}L\gamma_t^2 + L\gamma_t^2)\lambda_t^{2-\alpha}\sigma^\alpha$$

$$\leq \sum_{t=1}^{T-1} 36d_tL\gamma_t^2\lambda_t^{2-\alpha}\sigma^\alpha + \sum_{t=0}^{T-1} 108L\gamma_t^2\lambda_t^{2-\alpha}\sigma^\alpha, \tag{29}$$

where we use $d_{t+1} = d_t + 2\gamma_t L$, $L\gamma_t \leq 1$ and $d_0 = 0$. Next, we bound the first term from (29) as follows:

$$\sum_{t=1}^{T-1} 36d_tL\gamma_t^2\lambda_t^{2-\alpha}\sigma^\alpha \leq \sum_{t=1}^{T-1} 36d_{t+1}L\gamma_t^2\lambda_t^{2-\alpha}\sigma^\alpha \overset{(10)}{\leq} \sum_{t=1}^{T-1} \frac{36d_{t+1}^{\alpha/2}LR_0^{2-\alpha}\gamma_t^\alpha\sigma^\alpha}{80^{2-\alpha}\ln^{1-\alpha/2}\left(\frac{6(t+1)^2}{\delta}\right)}.$$

Subsequently, we can use (15) to bound $d_{t+1}$:

$$\sum_{t=1}^{T-1} 36d_tL\gamma_t^2\lambda_t^{2-\alpha}\sigma^\alpha \leq \sum_{t=1}^{T-1} \frac{36d_{t+1}^{\alpha/2}LR_0^{2-\alpha}\gamma_t^\alpha\sigma^\alpha}{80^{2-\alpha}\ln^{1-\alpha/2}\left(\frac{6(t+1)^2}{\delta}\right)}$$

$$\leq \sum_{t=1}^{T-1} \frac{36LR_0^{2-\alpha}\gamma_t^\alpha\sigma^\alpha}{80^{2-\alpha}\ln^{1-\alpha/2}\left(\frac{6(t+1)^2}{\delta}\right)}\cdot\left(\frac{6BL}{80\cdot13^{1/\alpha}}\frac{(t+1)^{1-\beta}-1}{1-\beta}\right)^{\alpha/2}.$$

Substitution of (9) with the boundary $B \leq \frac{R_0^{\frac{2}{3}}}{\sigma^{\frac{2}{3}}L^{\frac{1}{3}}}$ allows to derive

$$\sum_{t=1}^{T-1} 36d_tL\gamma_t^2\lambda_t^{2-\alpha}\sigma^\alpha \leq \sum_{t=1}^{T-1} \frac{36LR_0^{2-\alpha}\gamma_t^\alpha\sigma^\alpha}{80^{2-\alpha}\ln^{1-\alpha/2}\left(\frac{6(t+1)^2}{\delta}\right)}\cdot\left(\frac{6BL}{80\cdot13^{1/\alpha}}\frac{(t+1)^{1-\beta}-1}{1-\beta}\right)^{\alpha/2}$$

$$\leq \sum_{t=1}^{T-1} \frac{216LR_0^2}{80^{2-\alpha}(80\cdot13^{1/\alpha})^{\frac{3\alpha}{2}}\ln^{1+\alpha/2}\left(\frac{6(t+1)^2}{\delta}\right)}$$

$$\cdot\left(\frac{(t+1)^{1-\beta}-1}{1-\beta}\right)^{\alpha/2}(t+1)^{-\beta\alpha}.$$

Applying Lemma 3 to the RHS, we obtain

$$\sum_{t=1}^{T-1} 36 d_t L \gamma_t^2 \lambda_t^{2-\alpha} \sigma^\alpha \leq \sum_{t=1}^{T-1} \frac{216 L R_0^2}{80^{2-\alpha}(80 \cdot 13^{1/\alpha})^{\frac{3\alpha}{2}}(t+1)^{\beta\alpha-(1-\beta)\alpha/2}}. \tag{30}$$

To deal with the second term from (29), it is enough to substitute upper bounds on $\lambda_t$ and $\gamma_t$ with $B \leq \frac{R_0}{\sigma}$:

$$\sum_{t=0}^{T-1} 108 L \gamma_t^2 \lambda_t^{2-\alpha} \sigma^\alpha \overset{(10)}{\leq} \sum_{t=0}^{T-1} \frac{108 L R_0^{2-\alpha} \gamma_t^\alpha \sigma^\alpha}{80^{2-\alpha} \ln^{1-\alpha/2}\left(\frac{6(t+1)^2}{\delta}\right)} \overset{(9)}{\leq} \sum_{t=0}^{T-1} \frac{108 L R_0^2}{80^2 \cdot 13 (t+1)^{\beta\alpha}}. \tag{31}$$

Therefore, if we choose $\beta$ in the following way:

---

**Condition 6:**

$$\begin{cases} \beta\alpha \geq 1 \Rightarrow \beta \geq \frac{1}{\alpha}; \\ \beta\alpha - (1-\beta)\alpha/2 \geq 1 \Rightarrow \beta \geq \frac{2+\alpha}{3\alpha}, \end{cases} \tag{32}$$

---

combining (30), (31) and (32), one can obtain

$$⑥ \leq \sum_{t=0}^{T-1} \frac{216 L R_0^2}{80^{2-\alpha}(80 \cdot 13^{1/\alpha})^{\frac{3\alpha}{2}}(t+1)} + \sum_{t=0}^{T-1} \frac{108 L R_0^2}{80^2 \cdot 13(t+1)} \overset{(18)}{\leq} \frac{\Phi_0 \ln\left(\frac{6T^2}{\delta}\right)}{16},$$

where we also expand sum limits of the first term and apply (18).

**Bound for ⑦.** According to the event $E_{T-1}$, we have

$$\sum_{t=0}^{T-1} 2(d_{t+1} L \gamma_t^2 + L \gamma_t^2) \|\theta_t^b\|^2 \leq \sum_{t=0}^{T-1} \frac{2(d_{t+1} L \gamma_t^2 + L \gamma_t^2) \cdot 4^\alpha \sigma^{2\alpha}}{\lambda_t^{2\alpha-2}}$$

$$\overset{(10)}{\leq} \sum_{t=0}^{T-1} \frac{2 \cdot 80^{2\alpha-2}(d_t + 3) L \gamma_t^{2\alpha} \cdot 4^\alpha \sigma^{2\alpha} \ln^{\alpha-1}\left(\frac{6(t+1)^2}{\delta}\right)}{R_0^{2\alpha-2}} \cdot \max\{d_t, 1\}^{\alpha-1},$$

where in the last inequality we also use that $d_{t+1} = d_t + 2L\gamma_t \leq d_t + 2$. Next, if $d_t \leq 1$, we have that $d_t + 3 \leq 4$; otherwise, we get $d_t + 3 \leq 4d_t$. Thus, one can obtain $(d_t + 3)\max\{d_t, 1\}^{\alpha-1} \leq 4\max\{d_t^\alpha, 1\}$, which allows to have

$$\sum_{t=0}^{T-1} 2(d_{t+1} L \gamma_t^2 + L \gamma_t^2) \|\theta_t^b\|^2 \leq \sum_{t=1}^{T-1} \frac{8 \cdot 80^{2\alpha-2} d_t^\alpha L \gamma_t^{2\alpha} \cdot 4^\alpha \sigma^{2\alpha} \ln^{\alpha-1}\left(\frac{6(t+1)^2}{\delta}\right)}{R_0^{2\alpha-2}}$$

$$+ \sum_{t=0}^{T-1} \frac{8 \cdot 80^{2\alpha-2} L \gamma_t^{2\alpha} \cdot 4^\alpha \sigma^{2\alpha} \ln^{\alpha-1}\left(\frac{6(t+1)^2}{\delta}\right)}{R_0^{2\alpha-2}}, \tag{33}$$

where we apply $\max\{a, b\}^{\alpha-1} \leq a^{\alpha-1} + b^{\alpha-1}$ due to $\alpha - 1 \leq 1$ and combine terms according to the power of $d_t$ with mentioning that $d_0 = 0$ for the first term. The following idea is similar with previous bound – for the first term we use the bound on $d_t$ (15) and apply Lemma 3; after this, we substitute the choice of $\gamma_t$ with corresponding upper bound on $B$ for each term. Consequently, we

have

$$\sum_{t=1}^{T-1} \frac{8 \cdot 80^{2\alpha-2} d_t^\alpha L \gamma_t^{2\alpha} \cdot 4^\alpha \sigma^{2\alpha} \ln^{\alpha-1}\left(\frac{6(t+1)^2}{\delta}\right)}{R_0^{2\alpha-2}} \leq \sum_{t=1}^{T-1} \frac{8 \cdot 80^{2\alpha-2} d_{t+1}^\alpha L \gamma_t^{2\alpha} \cdot 4^\alpha \sigma^{2\alpha} \ln^{\alpha-1}\left(\frac{6(t+1)^2}{\delta}\right)}{R_0^{2\alpha-2}}$$

$$\overset{(15)}{\leq} \sum_{t=1}^{T-1} \frac{8 \cdot 80^{2\alpha-2} L \gamma_t^{2\alpha} \cdot 4^\alpha \sigma^{2\alpha} \ln^{\alpha-1}\left(\frac{6(t+1)^2}{\delta}\right)}{R_0^{2\alpha-2}} \cdot \left(\frac{6BL}{80 \cdot 13^{1/\alpha}} \frac{(t+1)^{1-\beta}-1}{1-\beta}\right)^\alpha$$

$$\overset{(9)}{\leq} \sum_{t=1}^{T-1} \frac{288 \cdot 80^{2\alpha-2} L R_0^2 \cdot 4^\alpha}{(80 \cdot 13^{1/\alpha})^{3\alpha} \ln^{\alpha+1}\left(\frac{6(t+1)^2}{\delta}\right)} \cdot \left(\frac{(t+1)^{1-\beta}-1}{1-\beta}\right)^\alpha (t+1)^{-2\beta\alpha}$$

$$\leq \sum_{t=1}^{T-1} \frac{288 \cdot 80^{2\alpha-2} L R_0^2 \cdot 4^\alpha}{(80 \cdot 13^{1/\alpha})^{3\alpha}(t+1)^{2\beta\alpha-(1-\beta)\alpha}}, \tag{34}$$

where we use the upper bound on $B$: $B \leq \frac{R_0^{\frac{2}{3}}}{\sigma^{\frac{2}{3}} L^{\frac{1}{3}}}$. For the second term from (33), we get

$$\sum_{t=0}^{T-1} \frac{8 \cdot 80^{2\alpha-2} L \gamma_t^{2\alpha} \cdot 4^\alpha \sigma^{2\alpha} \ln^{\alpha-1}\left(\frac{6(t+1)^2}{\delta}\right)}{R_0^{2\alpha-2}} \overset{(9)}{\leq} \sum_{t=0}^{T-1} \frac{8 \cdot 80^{2\alpha-2} \cdot 4^\alpha L R_0^2}{(80 \cdot 13^{1/\alpha})^{2\alpha}(t+1)^{2\beta\alpha}}, \tag{35}$$

where we apply $B \leq \frac{R}{\sigma}$. Hence, to obtain the sufficient bound, it is enough to choose $\beta$ as follows:

**Condition 7:**

$$\begin{cases} 2\beta\alpha \geq 1 \Rightarrow \beta \geq \frac{1}{2\alpha}; \\ 2\beta\alpha - (1-\beta)\alpha \geq 1 \Rightarrow \beta \geq \frac{1+\alpha}{3\alpha}; \end{cases} \tag{36}$$

As a result, combining (34), (35) and (36), we derive

$$⑦ \overset{(36)}{\leq} \sum_{t=1}^{T-1} \frac{288 \cdot 80^{2\alpha-2} 4^\alpha L R_0^2}{(80 \cdot 13^{1/\alpha})^{3\alpha}(t+1)} + \sum_{t=0}^{T-1} \frac{8 \cdot 80^{2\alpha-2} 4^\alpha L R_0^2}{(80 \cdot 13^{1/\alpha})^{2\alpha}(t+1)} \overset{(18)}{\leq} \frac{\Phi_0\left(\frac{6T^2}{\delta}\right)}{16},$$

where we abuse the notation $LR_0^2 = \Phi_0$.

**Final bound.** If we formulate events $E_①$, $E_③$, $E_⑤$ as follows:

$$E_① = \left\{ |①| \leq \frac{\Phi_0 \ln\left(\frac{6(T)^2}{\delta}\right)}{2} \text{ either } \sum_{t=0}^{T-1} \sigma_t^2 > \frac{\Phi_0^2 \ln\left(\frac{6(T)^2}{\delta}\right)}{24} \right\}$$

$$E_③ = \left\{ |③| \leq \frac{\Phi_0 \ln\left(\frac{6(T)^2}{\delta}\right)}{12} \text{ either } \sum_{t=0}^{T-1} \sigma_t^2 > \frac{\Phi_0^2 \ln\left(\frac{6(T)^2}{\delta}\right)}{1152} \right\}$$

$$E_⑤ = \left\{ |⑤| \leq \frac{\Phi_0 \ln\left(\frac{6(T)^2}{\delta}\right)}{8} \text{ either } \sum_{t=0}^{T-1} \sigma_t^2 > \frac{\Phi_0^2 \ln\left(\frac{6(T)^2}{\delta}\right)}{384} \right\},$$

then the event $E_{T-1} \cap E_① \cap E_③ \cap E_⑤$ implies

$$\Phi_T \leq \Phi_0 + \frac{\Phi_0 \ln\left(\frac{6T^2}{\delta}\right)}{2} + 3 \cdot \frac{\Phi_0 \ln\left(\frac{6T^2}{\delta}\right)}{12} + \frac{\Phi_0 \ln\left(\frac{6T^2}{\delta}\right)}{8} + 2 \cdot \frac{\Phi_0 \ln\left(\frac{6T^2}{\delta}\right)}{16}$$

$$\leq 2\Phi_0 \ln\left(\frac{6T^2}{\delta}\right).$$

Consequently, we have

$$\mathbb{P}\{E_T\} \geq \mathbb{P}\{E_{T-1} \cap E_① \cap E_③ \cap E_⑤\} = 1 - \mathbb{P}\{\overline{E}_{T-1} \cap \overline{E}_① \cap \overline{E}_③ \cap \overline{E}_⑤\}$$

$$\geq 1 - \mathbb{P}\{\overline{E}_{T-1}\} - \mathbb{P}\{\overline{E}_①\} - \mathbb{P}\{\overline{E}_③\} - \mathbb{P}\{\overline{E}_⑤\} \geq 1 - \delta \sum_{t=1}^{T-1} \frac{1}{t^2} - \frac{3\delta}{3T^2} = 1 - \delta \sum_{t=1}^{T} \frac{1}{t^2}.$$

This finishes the inductive step of the proof. Therefore, the event $E_K$ implies

$$\Phi_K \leq 2\Phi_0 \ln\left(\frac{6(K+1)^2}{\delta}\right)$$

with probability at least $1 - \delta \sum_{t=1}^{K} \frac{1}{t^2}$. Abusing the notation, with $d_0 = 0$ we have

$$d_K(f(x_K) - f^*) \leq 2LR_0^2 \ln\left(\frac{6(K+1)^2}{\delta}\right).$$

Thus, the event $E_K$ implies

$$f(x_K) - f^* \leq \frac{LR_0^2 \ln\left(\frac{6(K+1)^2}{\delta}\right)}{\sum_{t=0}^{K-1} \gamma_t L} = \frac{R_0^2 \ln\left(\frac{6(K+1)^2}{\delta}\right)}{\sum_{t=0}^{K-1} \gamma_t}. \tag{37}$$

Moreover, it is known that

$$\frac{1}{\sum_{i=1}^{n} \frac{1}{a_i}} \leq \frac{\sum_{i=1}^{n} a_i}{n^2}$$

for $a_i > 0$, since it is AM-HM inequality. Thus, applying it with $a_i = \frac{1}{\gamma_i}$ to (37) leads to

$$f(x_K) - f^* \leq \frac{R_0^2 \ln\left(\frac{6(K+1)^2}{\delta}\right)}{\sum_{t=0}^{K-1} \gamma_t} \leq \frac{R_0^2 \ln\left(\frac{6(K+1)^2}{\delta}\right)}{K^2} \sum_{t=0}^{K-1} \frac{1}{\gamma_t}.$$

Substituting (9), we have

$$f(x_K) - f^* \leq \frac{R_0^2 \ln\left(\frac{6(K+1)^2}{\delta}\right)}{K^2} \sum_{k=0}^{K-1} \frac{1}{\gamma_t}$$

$$\leq \frac{R_0^2 \ln\left(\frac{6(K+1)^2}{\delta}\right)}{K^2} \sum_{k=0}^{K-1} \max\left\{320L \ln\left(\frac{6(k+1)^2}{\delta}\right), \frac{80 \cdot 13^{1/\alpha}(k+1)^\beta \ln\left(\frac{6(k+1)^2}{\delta}\right)}{B}\right\}.$$

Before we provide a final bound, let us emphasize an important point. Combining **Conditions 1-7**, we should choose $\beta \geq \max\left\{\frac{1}{\alpha}, \frac{\alpha+1}{3\alpha-1}, \frac{2+\alpha}{3\alpha}, \frac{\alpha}{3\alpha-1}, \frac{\alpha+1}{3\alpha}\right\}$. It is easy to verify that $\beta$ should be greater than $\frac{2+\alpha}{3\alpha}$. What is more, using that

$$\ln\left(\frac{6(k+1)^2}{\delta}\right) \leq \ln\left(\frac{6(K+1)^2}{\delta}\right)$$

and

$$\sum_{k=0}^{K-1}(k+1)^\beta = \sum_{k=1}^{K} k^\beta \leq K^\beta + \int_1^K x^\beta dx = K^\beta + \frac{K^{1+\beta}-1}{1+\beta} \leq 2K^{1+\beta},$$

we conclude

$$f(x_K) - f^*$$

$$= \mathcal{O}\left(\frac{LR_0^2 \ln^2\left(\frac{6(K+1)^2}{\delta}\right)}{K} + \frac{\max\left\{R_0\sigma, L^{\frac{\alpha-1}{3\alpha-1}} R_0^{\frac{4\alpha-2}{3\alpha-1}} \sigma^{\frac{2\alpha}{3\alpha-1}}, L^{\frac{1}{3}} R_0^{\frac{4}{3}} \sigma^{\frac{2}{3}}\right\} \ln^2\left(\frac{6(K+1)^2}{\delta}\right)}{K^{1-\beta}}\right),$$

Noting that we choose $\beta$ as the best possible one, i.e. $\beta = \frac{2+\alpha}{3\alpha}$, we obtain

$$f(x_K) - f^* = \tilde{\mathcal{O}} \left( \frac{LR_0^2}{K} + \frac{\max\left\{ R_0\sigma, L^{\frac{\alpha-1}{3\alpha-1}} R_0^{\frac{4\alpha-2}{3\alpha-1}} \sigma^{\frac{2\alpha}{3\alpha-1}}, L^{\frac{1}{3}} R_0^{\frac{4}{3}} \sigma^{\frac{2}{3}} \right\}}{K^{\frac{2\alpha-2}{3\alpha}}} \right),$$

where $\tilde{\mathcal{O}}(\cdot)$ denotes polylogarithmic dependency. This concludes the proof. $\square$

