# OpenReview forum: "High-Probability Bounds for the Last Iterate of Clipped SGD"
_ICLR.cc/2026/Conference — ICLR 2026 Poster_

### Official Review · Reviewer_BKzV · 2025-10-25

**Soundness:** 2
**Presentation:** 3
**Contribution:** 1
**Rating:** 2
**Confidence:** 3

**Summary:**

The paper studies high-probability last-iterate guarantees for Clipped-SGD on convex L-smooth objectives under heavy-tailed noise with finite \alpha-th moments ($\alpha \in (1,2]$). It proposes a potential-based analysis with horizon-free step size and a clipping level scaling as $1/\sqrt{b_k}$, and proves a last-iterate rate $O(K^{-2(\alpha-1)/(3\alpha)})$.

**Strengths:**

- Addresses a practically relevant “last-iterate” question for clipped methods under heavy-tailed noise; the setup (convex + L-smooth + $\alpha$-moment) is standard and clearly stated.

- Writing is generally clear; related-work table situates results among expectation and high-probability bounds (SGD / clipped-SGD, average vs last iterate).

**Weaknesses:**

- Short and narrow in scope. The paper reads like a concise note; the empirical section is minimal (toy problems, limited baselines).
- Limited contribution vs. prior art. While the shift to high-probability \emph{last-iterate} guarantees under heavy tails is interesting, the improvement over existing results is incremental and tightly scoped to convex $L$-smooth objectives.
- Suboptimal rate at $\alpha=2$. Instantiating the main bound with $\alpha=2$ yields a last-iterate rate $\tilde{\mathcal{O}}(K^{-1/3})$, leaving a \emph{polynomial gap} to the known $\tilde{\mathcal{O}}(K^{-1/2})$ benchmark in the finite-variance setting.
- Overclaim / framing. Phrases such as “close a long-standing gap” and “first high-probability last-iterate guarantees” are stronger than warranted given the convex-only scope and the $\alpha=2$ suboptimality; related work is not contrasted with sufficient precision to delimit novelty.

**Questions:**

The paper is short with limited empirical support; the theoretical advance, while neat, is incremental and currently rate-suboptimal at $\alpha=2$; several claims read stronger than warranted. With strengthened results and a broader empirical study, it could be a good workshop paper.

---

> ### Author Response · Authors · 2025-11-20
> **Response to Reviewer BKzV**
>
> Dear Reviewer,
>
> Thank you for your time and feedback.
>
> > ***W1*** Short and narrow in scope. The paper reads like a concise note; the empirical section is minimal (toy problems, limited baselines).
>
> ***A:*** Brevity is not a weakness. We have addressed an important open question in the stochastic optimization literature about one of the most popular methods nowadays -- Clipped-SGD. Moreover, this is a theory paper, and our experiments are only ment to inllustrate our theoretical findings and motivation.
>
> > ***W2*** Limited contribution vs. prior art. While the shift to high-probability \emph{last-iterate} guarantees under heavy tails is interesting, the improvement over existing results is incremental and tightly scoped to convex-smooth objectives.
>
> ***A:*** Prior to this work there was no existing high-probability rate for the last iterate of Clipped-SGD, in this regard we see our contribution as substantial. The case of convex-smooth objective is foundational and unquestioned in stocahstic optimization.
>
> > ***W3*** Suboptimal rate at $\alpha = 2$.
>
> ***A:*** Thank you for raising this important point. We agree that in the finite-variance case ($\alpha = 2$) our high-probability bound for the last iterate of clipped SGD exhibits a polynomial gap compared to the optimal $O(1/\sqrt{K})$ expectation rate. We already acknowledge this limitation in the paper. However, this does not diminish the contribution for several reasons.
>
> First, our setting is substantially more challenging: we aim at *high-probability* last-iterate guarantees for *Сlipped*-SGD under a finite-moment noise model. To the best of our knowledge, no such bounds were previously available, even in the basic smooth and convex setting. Importantly, it is not known whether the optimal expectation rate $O(1/\sqrt{K})$ can be upgraded to a high-probability guarantee with only $\mathrm{polylog}(1/\delta)$ dependence on the failure probability $\delta$. In this sense, the gap in our result — our rate scales as $O(K^{-1/3})$ (ignoring additional parameters) — does not contradict any known theory, and closing it remains an open problem.
>
> Second, even in the standard *unclipped* case with finite variance, sharp last-iterate analyses are extremely recent. The optimal last-iterate rate $O(1/\sqrt{K})$ for smooth convex objectives was established only recently (Liu \& Zhou, 2023), while for nearly a decade the state of the art was the $O(K^{-1/3})$ rate of Bach \& Moulines (2014). Likewise, almost-sure last-iterate convergence rates were obtained only in very recent work (Li 2022). In this context, our result provides, to the best of our knowledge, the first high-probability last-iterate guarantee for Сlipped-SGD under heavy-tailed noise.
>
> For these reasons, we believe that although the bound is not sharp when $\alpha = 2$, it still represents a meaningful advance and opens several promising directions for future work—including the possibility of matching the optimal $O(1/\sqrt{K})$ rate in high probability.
>
> > ***W4*** Overclaim / framing.
>
> ***A:*** Since its rise in popularity around 2020, **no results on the last iterate of Clipped-SGD** have been established in the settings considered in our paper. In modern optimization, a five-year gap is substantial, so it is reasonable to refer to the addressed question as **"long-standing."**
>
> Moreover, the phrase **"first high-probability last-iterate guarantees"** accurately describes our contribution -- this is a factual statement, not an overclaim. Finally, related works are properly discussed, as we cite all existing high-probability bounds for Clipped-SGD within the considered setting.
>
> Taking these aspects into account, we believe our **positioning within the literature is appropriate and well justified.**

---

> > ### Author Response · Authors · 2025-11-27
> >
> > Dear Reviewer BKzV,
> >
> > We would like to kindly follow up on our rebuttal to your review. We carefully addressed all the points raised in your review and would greatly appreciate knowing whether our response clarified your concerns.
> >
> > If any issues remain, we would be happy to provide further details.
> >
> > Thank you for your time and consideration.
> >
> > Best regards,
> >
> > Authors

---

### Official Review · Reviewer_TuTb · 2025-10-26

**Soundness:** 2
**Presentation:** 2
**Contribution:** 2
**Rating:** 6
**Confidence:** 4

**Summary:**

The paper provides the first high-probability last-iterate convergence rate of Clipped SGD. The authors also conducted numerical experiments demonstrating that the last iterate of Clipped SGD indeed converges.

**Strengths:**

The result is meaningful, as it fills a gap in the literature.

**Weaknesses:**

1. Line 112, when talking about the heavy-tailed noise, the finite-sum problem is not a proper example. Note that for this kind of problem, a finite $\alpha$-th central moment directly implies a finite $2$nd central moment.

1. Line 116, missing $\\\{\\\}$ for ${x_k}_{k=0}^K$.

1. Some statements in Section 3 are inaccurate.

    1. Line 142, $\widetilde{\mathcal{O}}$ is not necessary and instead should be $\mathcal{O}$ since the current description doesn't limit to the case of an unknown $K$.

    1. Line 149, missing $)$ after $\mathbb{E}[...]$. In addition, Fatkhullin et al. (2025) didn't consider exactly the same condition as in the current Assumption 3.

    1. For the in-expectation convergence under heavy-tailed noise, the authors missed a relevant work [1]. Note that for the non-smooth Lipschitz case, [1] showed both average-iterate and last-iterate convergence. For the smooth case, [1] also showed the average-iterate convergence.

    1. Still in Table 1, if the results in (Nguyen et al., 2023) are better than (Sadiev et al., 2023) from every perspective, then I don't see the point of putting (Sadiev et al., 2023) in it.

    1. Line 192,  none of these four works (Nazin et al., 2019; Gorbunov et al., 2020; Gorbunov et al., 2024; Parletta et al., 2024) consider heavy-tailed noise. (Liu & Zhou, 2024) didn't study clipping methods.

    **Reference**

    [1] Liu, Z. (2025). Online Convex Optimization with Heavy Tails: Old Algorithms, New Regrets, and Applications. arXiv preprint arXiv:2508.07473.

1. In the statement of Theorem 1:

    1. No definition of $d_k$ is provided.

    1. I don't see why $p$ is a function of $L$. Do the authors mean $C$?

    1. The definition of $C$ is also not proper. It involves solving a fixed-point equation. However, the authors didn't discuss whether it is always solvable (or any other related discussion). In addition, since $\Phi_0=C\\\|x_0-x^\*\\\|^2/2$, which cannot be evaluated in general due to the unknown value $\\\|x_0-x^*\\\|$.

1. The whole paper relies on the existence of $x^*$, which the authors should state clearly.

1. About numerical experiments:

    1. For all experiments, could the authors also provide the empirical mean and standard deviation for both the average iterate and the last iterate?

    1. In Line 307, the authors said the step-size and clipping level schedules are selected based on the theory. Could the authors provide more details? Especially, for $f(x)=\ln(1+\exp(\langle x,a\rangle))$, this objective doesn't have an optimal solution on $\mathbb{R}^d$ (also implying $\\\|x_0-x^\*\\\|=+\infty$), which clearly differs from the current theorem. Could the authors provide more details?

1. I appreciate the authors' honesty about the limitations discussed in Section 6. However, I have to repeat that the current rate $\tilde{O}(1/K^\frac{2(\alpha-1)}{3\alpha})$ seems not optimal.

1. The analysis is somewhat standard, without introducing any new techniques, as far as I can see.

1. The proof is not written in a mathematically rigorous manner. For example, the two inequalities in Lines 893-897 only hold conditionally on the event $E_{T-1}$. This means that one cannot directly invoke Bernstein’s inequality in the following proof. Instead, one should do one more step to introduce some proxy variables, like many prior works (e.g., the indicator variable $\mathbb{I}[\Phi_t\leq 2\Phi_0\log(6(t+1)^2/\delta)]$). Therefore, I believe many parts of the proof should be carefully revised.

**Questions:**

In addition to the weaknesses mentioned earlier, it seems the current proof cannot be directly extended to constrained optimization. Could the authors provide any discussion on this issue?

---

> ### Author Response · Authors · 2025-11-20
> **Response to Reviewer TuTb**
>
> Dear Reviewer,
>
> Thank you for your time and feedback.
>
> > ***W1*** Line 112, when talking about the heavy-tailed noise, the finite-sum problem is not a proper example. Note that for this kind of problem, a finite $\alpha$-th central moment directly implies a finite 2nd central moment.
>
> ***A:*** We thank the reviewer for pointing out that, under *uniform sampling* in a finite-sum problem, the stochastic gradient takes only finitely many values and therefore has all finite moments. This observation is correct in a strict formal sense. However, our intention was not to claim that uniform finite-sum sampling inherently produces heavy-tailed noise, but rather to illustrate the structure of the stochastic gradient oracle.
>
> More importantly, boundedness of the stochastic gradient in the finite-sum setting does *not* imply that a high-probability analysis becomes easier or that the heavy-tailed model is unnecessary. High-probability bounds based purely on the gradient *range* can be significantly worse than bounds based on finite $\alpha$-th moments, especially when the range is large (e.g., due to unnormalized features, heterogeneous component functions, or the presence of outliers). In many practical and theoretical settings, the range-based constants are far too large to yield meaningful high-probability guarantees, even though all moments are technically finite.
>
> Furthermore, the tail behavior and variance of the stochastic gradient depend on the *sampling strategy*, not only on the fact that the problem is a finite sum. Importance sampling or other non-uniform sampling schemes can lead to heavy-tailed estimators even when the underlying loss components are bounded. Our finite-$\alpha$-moment noise model is designed to capture precisely these situations and is substantially more general and realistic than assuming a uniform bounded range.
>
> For these reasons, we agree that the uniform finite-sum case is not a canonical example of heavy-tailed noise. Nonetheless, the heavy-tailed model and our analysis remain highly relevant even in finite-sum problems whenever range-based bounds are too loose or when non-uniform sampling is used.
>
> > ***W2:*** Line 116, missing $\lbrace \rbrace$
>
> ***A:*** Thank you, the typo is fixed in the revised version (all changes are highlighted in orange).
>
> > ***W3:*** Some statements in Section 3 are inaccurate.
>
> >> ***W3-1:*** Line 142, $\widetilde{O}$ is not necessary
>
> ***A:*** Thank you, the typo is fixed.
>
> >> ***W3-2:*** Line 149, missing $)$ after $\mathbb{E}[...]$. In addition, Fatkhullin et al. (2025) didn't consider exactly the same condition as in the current Assumption 3.
>
> ***A:*** Thank you, the typo is fixed and the text is refined.
>
> >> ***W3-3:*** For the in-expectation convergence under heavy-tailed noise, the authors missed a relevant work [1]. Note that for the non-smooth Lipschitz case, [1] showed both average-iterate and last-iterate convergence. For the smooth case, [1] also showed the average-iterate convergence.
>
> ***A:*** Thank you for pointing out this work. We have added the discussion of the results from this paper to footnote 2 of the revised version.
>
> >> ***W3-4:*** Still in Table 1, if the results in (Nguyen et al., 2023) are better than (Sadiev et al., 2023) from every perspective, then I don't see the point of putting (Sadiev et al., 2023) in it.
>
> ***A:*** There is a small yet important difference between two results: as indicated in the table, Sadiev et al. (2023) make all the assumptions only on some ball around the solution, while Nguyen et al. (2023) make all the assumptions globally.
>
> >> ***W3-5:*** Line 192, none of these four works (Nazin et al., 2019; Gorbunov et al., 2020; Gorbunov et al., 2024; Parletta et al., 2024) consider heavy-tailed noise. (Liu & Zhou, 2024) didn't study clipping methods.
>
> ***A:*** We clarified this part in footnote 3.
>
> Please refer to the following official comment for the notes on W4 - W9 and Q1.

---

> > ### Author Response · Authors · 2025-11-20
> > **Continuation of the response to Reviewer TuTb**
> >
> > This comment is a continuation of the previous one, and here we respond to W4, W5 and W6.
> >
> > > ***W4:*** In the statement of Theorem 1
> >
> > >> ***W4-1:*** No definition of $d_k$ is provided
> >
> > ***A:*** Thank you, we have added the definition of $d_k$ to the final version.
> >
> > >> ***W4-2:*** I don't see why $p$ is a function of $L$. Do the authors mean $C$?
> >
> > ***A:*** Thank you for pointing this out. In the original version, we clarified this part in Remark 4 (Appendix B.3). To improve readability, we have added an explicit formula for $p(\Phi_0, L, \sigma)$ to the revised statement of Theorem 1.
> >
> > >> ***W4-3:*** The definition of $C$ is also not proper.
> >
> > ***A:*** We discuss the definition of $p(\Phi_0, L, \sigma)$ and $C$ in Appendix B.3, see Remark 4 and formulas (26) and (27) in the revised version (formulas (25) and (26) in the original version).
> >
> > > ***W5:*** The whole paper relies on the existence of x_*, which the authors should state clearly.
> >
> > ***A:*** Thank you for this comment. Yes, indeed, we implicitly assume the existence of an optimal point $x^\ast$. We have included this assumption in the revised version.
> >
> > > ***W6*** About numerical experiments:
> >
> > >> ***W6-1:*** For all experiments, could the authors also provide the empirical mean and standard deviation for both the average iterate and the last iterate?
> >
> > ***A:*** Thank you. In the appendix of the revised manuscript we will add a figure reporting, the mean and the standard deviation of the errors, in addition to the percentiles.
> >
> > >> ***W6-2:*** In Line 307, the authors said the step-size and clipping level schedules are selected based on the theory. Could the authors provide more details? Especially, for f(x)=\ln(1+\exp(\langle x, a\rangle)), this objective doesn't have an optimal solution on $\mathbb{R}^d$ (also implying $\|x_0-x_*\|= \infty$), which clearly differs from the current theorem. Could the authors provide more details?
> >
> > ***A:*** We thanks the reviewer for pointing this out.
> >
> > For the step-size and the clipping level, we resort to the schedules correspoding to the case $\alpha=2$, i.e., to $\gamma_k = \frac{c_1}{(k+1)^{2/3} \log^3(k)}$ and $\lambda_k = c_2 (k+1)^{2/3} \sqrt{\log(k)}$, and optimize the constants $c_1$ and $c_2$ via grid-search, individually, for the average and the last iterate.
> >
> > The reviewer is right in pointing out that for the logistic regression problem, the infimum $f_\ast=0$ is not achieved. Indeed the solution of Clipped-SGD will converge to a vector aligned with $a$ and with unbounded norm. In that case, we are merely interested in analyzing the decay of the optimization error, even if the setting is not exactly matching the specification that $\arg \min f \neq \emptyset$.
> >
> > Please refer to the following official comment for the notes on W7 - W9 and Q1.

---

> > > ### Author Response · Authors · 2025-11-20
> > > **Continuation of the response to Reviewer TuTb**
> > >
> > > This comment is a continuation of the previous one, and here we respond to W7, W8, W9 and Q1.
> > >
> > > >***W7*** I appreciate the authors' honesty about the limitations discussed in Section 6. However, I have to repeat that the current rate $\tilde{\mathcal{O}}\bigg(\frac{1}{K^{\frac{2(\alpha - 1)}{3\alpha}}}\bigg)$ seems not optimal.
> > >
> > > ***A:*** Thank you for raising this important point. We agree that in the finite-variance case ($\alpha = 2$) our high-probability bound for the last iterate of clipped SGD exhibits a polynomial gap compared to the optimal $O(1/\sqrt{K})$ expectation rate. We already acknowledge this limitation in the paper. However, this does not diminish the contribution for several reasons.
> > >
> > > First, our setting is substantially more challenging: we aim at *high-probability* last-iterate guarantees for *Сlipped*-SGD under a finite-moment noise model. To the best of our knowledge, no such bounds were previously available, even in the basic smooth and convex setting. Importantly, it is not known whether the optimal expectation rate $O(1/\sqrt{K})$ can be upgraded to a high-probability guarantee with only $\mathrm{polylog}(1/\delta)$ dependence on the failure probability $\delta$. In this sense, the gap in our result — our rate scales as $O(K^{-1/3})$ (ignoring additional parameters) — does not contradict any known theory, and closing it remains an open problem.
> > >
> > > Second, even in the standard *unclipped* case with finite variance, sharp last-iterate analyses are extremely recent. The optimal last-iterate rate $O(1/\sqrt{K})$ for smooth convex objectives was established only recently (Liu \& Zhou, 2023), while for nearly a decade the state of the art was the $O(K^{-1/3})$ rate of Bach \& Moulines (2014). Likewise, almost-sure last-iterate convergence rates were obtained only in very recent work (Li 2022). In this context, our result provides, to the best of our knowledge, the first high-probability last-iterate guarantee for Сlipped-SGD under heavy-tailed noise.
> > >
> > > For these reasons, we believe that although the bound is not sharp when $\alpha = 2$, it still represents a meaningful advance and opens several promising directions for future work—including the possibility of matching the optimal $O(1/\sqrt{K})$ rate in high probability.
> > >
> > > >***W8*** The analysis is somewhat standard, without introducing any new techniques, as far as I can see.
> > >
> > > ***A*** Unfortunately, we kindly disagree with this statement. According to the existing literature, this analysis is the first to establish high-probability convergence for the last iterate of Clipped-SGD. The technical novelty lies at least in the horizon-free parameter choices: under heavy-tailed stochasticity, this is the first result that achieves high-probability convergence without dependence on the total number of iterations. We understand that the result may appear standard, and this is natural -- every analysis builds upon some foundational framework. Nevertheless, we believe it would be unfair to call this work standard: it was developed for the first time, and several technical components -- such as independence from the total iteration count and the use of a decreasing clipping level in the convex setting -- have not appeared in prior literature.
> > >
> > > >***W9*** The proof is not written in a mathematically rigorous manner. For example, the two inequalities in Lines 893-897 only hold conditionally on the event $E_{T-1}$. This means that one cannot directly invoke Bernstein’s inequality in the following proof. Instead, one should do one more step to introduce some proxy variables, like many prior works (e.g., the indicator variable). Therefore, I believe many parts of the proof should be carefully revised.
> > >
> > > ***A:*** Thank you very much for this very important remark. You are absolutely right — for mathematical correctness, one must use truncated versions of the variables; otherwise, applying concentration inequalities is not fully rigorous from a formal standpoint. Following an approach similar to (Sadiev et al., 2023), we introduced additional variables $\eta_t$ and $\mu_t$ as truncated versions of $\nabla f(x_t)$ and $x_t - x^\ast$, respectively. With these, the concentration inequality can be applied properly. We also double-checked the proof. Thank you again -- ensuring the correctness of the proof is extremely important. We have highlighted the changes in the proof in orange.
> > >
> > > > ***Q1:*** In addition to the weaknesses mentioned earlier, it seems the current proof cannot be directly extended to constrained optimization. Could the authors provide any discussion on this issue?
> > >
> > > ***A:*** Thank you for the very interesting question. To the best of our knowledge, there are no potential-based proofs for Projected SGD in the literature even under light-tailed noise assumption. Therefore, extending our analysis to the constrained optimization is not trivial and requires a separate study.

---

> > > > ### Comment · Reviewer_TuTb · 2025-11-28
> > > >
> > > > Dear Authors,
> > > >
> > > > Thanks for your detailed reply. After reading the rebuttal, I decided to maintain my current score. As stated in the original review, I think the result itself is meaningful. However, I cannot provide a higher score, since the current bound is likely sub-optimal and, at least from my perspective, introduces no new technical insights.
> > > >
> > > > In the following, I will reply to the authors' comments.
> > > >
> > > >  **To A1:** For the finite-sum problem, even for importance sampling (I assume the authors refer to sample $\nabla f_i/p_i$ with probability $p_i$), a finite $\alpha$-th moment on the noise still implies a finite $2$nd moment on the noise (in fact, any $m$-th moment exists). Of course, the moment itself could depend on the sampling strategy and $n$. But the importance is that a finite $m$-th moment (for simplicity, let's say $m>4$) provably yields a last-iterate rate $1/\sqrt{T}$ (for large enough $T$). For example, see [1]. I understand the settings considered in [1] and the current paper are not identical. Here, I only want to say that the finite-sum problem could possibly lead to a better rate and, again, is not a proper example for the model studied in the current paper.
> > > >
> > > > **Reference**
> > > >
> > > > [1] Eldowa, Khaled, and Andrea Paudice. "General tail bounds for non-smooth stochastic mirror descent." International Conference on Artificial Intelligence and Statistics. PMLR, 2024.
> > > >
> > > >  **To A3-4:** The reply is not convincing. Note that the proof of Nguyen et al. (2023) essentially stated a fact $\\\|x_{k+1}-x^\*\\\|\leq 2\\\|x\_1-x^*\\\|$ (see the last inequality on their Page 26). As such, their assumptions only need to hold for a small ball, as in Sadiev et al. (2023).
> > > >
> > > > **To A4-2&3:** Thanks for the clarification.
> > > >
> > > > **To A6-1:** I didn't find the new reports. Did I miss anything?
> > > >
> > > > **Additional comment:** This is only a suggestion that won't change my rating: the authors could try $\alpha\neq 2$ to make the paper more compelling.
> > > >
> > > > **To A7:** As mentioned, I think this is a meaningful result, but there may be a large room for improvement.
> > > >
> > > > **To A8:** The statement is misleading. For horizon-free parameter choices, see Theorem 4.4 of Nguyen et al. (2023).
> > > >
> > > > **To A9:** Line 1006, I think $\\\|x_t\\\|$ should be $\\\|x_t-x^*\\\|$.
> > > >
> > > > **To Q&A:** Thanks for the answer. I encourage the authors to explore the possibility in the future.
> > > >
> > > > Best regards,
> > > >
> > > > Reviewer

---

> > > > > ### Author Response · Authors · 2025-12-01
> > > > >
> > > > > We thank the reviewer for engaging with our rebuttal. Below we provide further clarifications.
> > > > >
> > > > > ***1.***
> > > > > We agree that the finite-sum setting is not the main target application of our results. However, we would like to highlight an important difference with respect to the results in [1].
> > > > >
> > > > > First, the analysis in [1] focuses on the case $p > 4$, which is indeed a stronger moment assumption than the one considered in our work, and on Lipschitz but possibly non-smooth objectives (while here we assume smoothness). Nevertheless, for the \emph{last iterate}, the rates proved in [1] are of the order (see Corollary 3(ii))
> > > > >
> > > > > $\frac{1}{\delta^{2/p}\sqrt{k}}$
> > > > >
> > > > > (ignoring logarithmic factors in $k$), while the improved rates (i.e. quick decay of the $poly(1/\delta)$ term) are only achied for the \emph{averaged iterate}. In contrast, our results provide a *polylogarithmic dependence* on $1/\delta$ for the last iterate, which is significantly better than the polynomial dependence appearing in [1].
> > > > >
> > > > > We also note that even in the classical case $p=2$, a rate of order $(1/\delta)/\sqrt{k}$ can be straightforwardly obtained from standard in-expectation convergence guarantees combined with Markov’s inequality. However, such bounds suffer from a very poor dependence on the failure probability. Indeed, the main advantage of clipping in our analysis lies precisely in ensuring a *polylogarithmic dependence* on $1/\delta$.
> > > > >
> > > > > To better quantify this advantage, consider for instance a moderately small failure probability, say $\delta = 10^{-3}$. In this case, the bound derived via Markov’s inequality is already a factor $10^{3}$ worse than its in-expectation counterpart. On the other hand, our results guarantee a bound of the order $81/k^{1/3}$ through clipping. This bound is tighter than the one obtained via Markov’s inequality for all
> > > > >
> > > > > $k \leq 3 \times 10^{6}$
> > > > >
> > > > > Illustrating the practical relevance of the improved $\delta$-dependence achieved by our approach.
> > > > >
> > > > > ***2. - 8.***
> > > > > We agree with the reviewer and removed *Sadiev et al.* from the state-of-the-art table.
> > > > >
> > > > > ***3.***
> > > > > We thank the reviewer for pointing this out. We have revised the experimental section (see the blue text in the revised manuscript) to incorporate all of the reviewer’s suggestions. In particular, we have calibrated the logistic regression experiments by adding a small regularization term to ensure the problems are well-defined, and we now report the sample mean and standard deviation, together with the 0.95–quantile (which is more representative in our setting), in all figures.
> > > > >
> > > > > ***4.***
> > > > > We are unsure whether we fully understand the reviewer’s suggestion regarding the case $\alpha \neq 2$, since our general results already hold for all $\alpha \in (1,2]$. We would appreciate further clarification.

---

### Official Review · Reviewer_7EUw · 2025-10-31

**Soundness:** 4
**Presentation:** 4
**Contribution:** 2
**Rating:** 2
**Confidence:** 4

**Summary:**

This paper presents an analysis of SGD for heavy-tailed gradients with bounded alpha-moment and convergence rate 1/T^{2(alpha-1)/(3alpha)} for smooth convex objectives.

**Strengths:**

I am not aware of other results on this problem, and the execution seems good.

**Weaknesses:**

The convergence rate is not optimal. This makes the paper somewhat incremental because if you allow projection into a bounded diameter domain, you get the optimal rate (without clipping). This is a straightforward consequence of recent results connecting the last iterate of SGD with linear decay schedules and regret analysis, along with the same martingale/bias arguments used to bound error from clipping in this paper (see e.g. https://arxiv.org/abs/2310.07831 and the more general result in https://arxiv.org/abs/2405.15682). Essentially, one needs only show that the linearized regret of projected online gradient descent is bounded in high probability, which I think is not too hard using the standard concentration of clipped values arguments presented here if the gradients are clipped and the iterates are bounded by some projection.

Without the projection, there is some difficulty and so the present paper does have value. My concern is that this difference seems relatively small.

If it were possible to achieve the optimal rate without projection, or (even better) show that the optimal rate is NOT achievable without projections, that would be a much better result.

**Questions:**

Please address the weakness above. Happy to revise my opinion.

---

> ### Author Response · Authors · 2025-11-20
> **Response to Reviewer 7EUw**
>
> Dear Reviewer,
>
> Thank you for your time and feedback.
>
> >***W1*** The convergence rate is not optimal. This makes the paper somewhat incremental because if you allow projection into a bounded diameter domain, you get the optimal rate (without clipping). This is a straightforward consequence of recent results connecting the last iterate of SGD with linear decay schedules and regret analysis, along with the same martingale/bias arguments used to bound error from clipping in this paper (see e.g. https://arxiv.org/abs/2310.07831 and the more general result in https://arxiv.org/abs/2405.15682). Essentially, one needs only show that the linearized regret of projected online gradient descent is bounded in high probability, which I think is not too hard using the standard concentration of clipped values arguments presented here if the gradients are clipped and the iterates are bounded by some projection.
>
> >Without the projection, there is some difficulty and so the present paper does have value. My concern is that this difference seems relatively small.
>
> >If it were possible to achieve the optimal rate without projection, or (even better) show that the optimal rate is NOT achievable without projections, that would be a much better result.
>
> ***A:*** We thank the reviewer for the detailed discussion and for pointing out the connection to regret-based analyses of last-iterate SGD. However, we believe the comparison somewhat overstates how *straightforward* it is to obtain our type of result from the cited works.
>
> - The references suggested by the reviewer do not analyze *Clipped*-SGD under heavy-tailed noise, and (to the best of our understanding) their guarantees are in *expectation*, not in high probability for the last iterate. In contrast, our goal is to provide *high-probability* last-iterate guarantees for *Clipped*-SGD under a finite $\alpha$-th moment noise model. This is a substantially different setting, and the results in the cited papers do not directly imply our bounds.
>
> - The claim "*if you allow projection into a bounded diameter domain, you get the optimal rate (without clipping)*" in the heavy-tailed, clipped setting is not backed by a formal proof in the literature, as far as we are aware. Projecting the iterates onto a bounded domain does not remove heavy tails of the stochastic gradient: even if $||x_k - x_\ast||$ is a priori bounded by projection, the noise term in the gradient estimator remains heavy-tailed. Bridging from *expected* regret bounds for projected online gradient descent to *high-probability* last-iterate guarantees for *Clipped*-SGD under heavy-tailed noise is, in our view, a nontrivial technical step and certainly not a one-line consequence of existing results.
>
> - It is well understood in the stochastic optimization literature that removing the bounded-domain assumption is significantly more challenging (e.g., Gorbunov et al. 2020, Li et al. 2023, Nguyen et al. 2024, Eldowa & Paudice 2024, Liu & Zhou 2024). Without projections, one cannot rely on a uniform bound of the form $||x_k - x_\ast|| \leq D$ and must instead control quantities like $\sum_{k=1}^K \langle g_k, x_k - x_\ast\rangle$, where $g_k$ are (clipped/heavy-tailed) gradient estimators and the distribution of $x_k - x_\ast$ itself is nontrivial to characterize. Our analysis is specifically designed to handle this unbounded-domain regime, which requires different tools than those used in standard regret proofs.
>
> Therefore, our contribution are not a minor variant of a *projection-based* argument. Rather, we provide, to the best of our knowledge, the first high-probability last-iterate guarantees for Clipped-SGD under finite-moment noise *without* assuming bounded domains. We agree that obtaining optimal rates in this setting, or proving that they are unattainable without projections, would be highly interesting; we see this as a natural and technically demanding direction for future work, rather than something that follows directly from current regret-based analyses.

---

> > ### Author Response · Authors · 2025-11-27
> >
> > Dear Reviewer 7EUw,
> >
> > In your review, you mentioned that you would consider increasing your score if our rebuttal addressed your concerns. We therefore wanted to kindly follow up, as we have not yet seen a response to our rebuttal and would be happy to further clarify any remaining doubts.
> >
> > Thank you again for your time and for engaging with our work.
> >
> > Best regards,
> >
> > Authors

---

### Official Review · Reviewer_Mpxc · 2025-11-01

**Soundness:** 3
**Presentation:** 3
**Contribution:** 3
**Rating:** 2
**Confidence:** 3

**Summary:**

This paper studies **Clipped Stochastic Gradient Descent (Clipped-SGD)** under **heavy-tailed noise**—that is, when the stochastic gradient only has a finite α-th moment for α ∈ (1, 2]. The authors provide, for the **first time**, *high-probability* convergence guarantees for the **last iterate** (as opposed to the more typical average iterate) on convex, L-smooth objectives.

Their main result shows that the last iterate of Clipped-SGD achieves a convergence rate of
[
O!\left(\frac{\mathrm{polylog}(K/\delta)}{K^{2(\alpha-1)/(3\alpha)}}\right)
]
with failure probability ≤ δ.

They also propose *horizon-free* step-size (γₖ) and clipping-level (λₖ) schedules—meaning they do not require knowing the total number of iterations in advance—and verify their theory empirically.

**Strengths:**

1. **First High-Probability Last-Iterate Result under Heavy Tails**
   Previous work established only *expected* convergence or *average-iterate* guarantees. This paper closes a notable theoretical gap by analyzing the **last iterate** under α-th moment assumptions, a more realistic model for gradient noise in large-scale ML.

2. **Novel Potential-Based Proof**
   The authors design a custom potential function Φₖ combining function suboptimality and distance to optimum. This enables applying **Freedman/Bernstein inequalities** for martingale control, producing high-probability results without strong boundedness assumptions.

3. **Horizon-Free (Any-Time) Step-Size and Clipping**
   The method adaptively scales γₖ and λₖ using logarithmic corrections in δ, ensuring convergence without pre-specifying total iterations. This is practical for streaming or online settings.

4. **Solid Empirical Verification**
   The authors validate theory with synthetic and logistic regression problems under heavy-tailed (Pareto, Student-t) noise, showing that the **last iterate consistently outperforms the average iterate**.

**Weaknesses:**

1. **Limited Empirical Scope**
   Only simple convex problems (logistic, quadratic) are tested. Including real datasets or large-scale ML examples (e.g., transformer finetuning) would strengthen impact.

2. **Complex Presentation**
   The notation (γₖ, λₖ, dₖ, bₖ, Φ₀, C, p(Φ₀,L,σ)) is extremely dense. The exposition would benefit from a simplified summary table for parameters and clearer intuition for each scaling choice.

3. **Lack of Sharpness at α=2**
   When α = 2 (finite variance), the bound leaves a polynomial gap from the optimal O(1/√K) rate. The authors acknowledge this, but it remains an open issue.

4. **δ-Dependence Obscures Practical Interpretation**
   The interplay between the learning rate γ and the probability parameter δ—key to your question—is not explicitly analyzed beyond asymptotic scaling.

**Questions:**

In your final convergence result, the learning rate $\gamma$ explicitly depends on the confidence parameter $\delta,$ i.e., $\gamma_k \propto 1 / \ln^3(1/\delta).$
This dependence between the step size and the high-probability confidence level is quite unusual in the stochastic optimization literature, where $\gamma$ is typically independent of probabilistic guarantees.
Could the authors clarify **why this dependence is necessary** in your analysis?
Is it an intrinsic requirement of the heavy-tailed, high-probability setting, or merely a technical artifact introduced by the proof method (e.g., via union bounds or Freedman’s inequality)?

If this coupling is indeed required, the authors should provide a **rigorous necessity proof** to justify it, since this assumption significantly restricts the generality of the result.
 Moreover, **if such δ-dependent compression of the learning rate is generally allowed**, then even many divergent dynamical systems — for example,
$$
x_{n+1} = x_n - \gamma_n x_n^2+\gamma_n x_n^4 + \gamma_n z_n, \quad (z_n \sim \mathcal{N}(0,1))
$$
—could be made to appear *artificially stable* or “high-probability convergent” simply by shrinking $\gamma_n$ according to a small δ.
This type of construction is **well known from standard graduate-level stochastic process exercises taught in mathematics departments**, and therefore **cannot be regarded as a publishable theoretical contribution** unless the authors rigorously prove that the δ-dependence is both mathematically necessary and not a by-product of overly conservative probabilistic bounding

---

> ### Author Response · Authors · 2025-11-20
> **Response to Reviewer Mpxc**
>
> Dear Reviewer,
>
> Thank you for your time and feedback.
>
> > ***W1:*** **Limited Empirical Scope**, Only simple convex problems (logistic, quadratic) are tested. Including real datasets or large-scale ML examples (e.g., transformer finetuning) would strengthen impact.
>
> ***A:*** Thank you for the helpful comment. Our primary contribution is theoretical: we characterize the behavior of the last iterate and provide non-asymptotic guarantees. The experiments are included to complement our theoretical bounds and suggest that they might be improved, though **no lower bounds for the last iterate of Clipped-SGD are known**. For this reason, we focus on settings that are fully aligned with our theoretical assumptions, namely convex and smooth objectives.
>
> Extending the empirical evaluation to large-scale or non-convex problems (e.g., fine-tuning transformers) is indeed interesting, but goes beyond the scope of the theoretical framework developed in the paper: our guarantees do not apply in those regimes, and including such experiments could be misleading regarding the formal claims. We have therefore chosen datasets and models that faithfully match the setting studied in our analysis.
>
> >***W2:*** **Complex Presentation**. The notation (γₖ, λₖ, dₖ, bₖ, Φ₀, C, p(Φ₀,L,σ)) is extremely dense. The exposition would benefit from a simplified summary table for parameters and clearer intuition for each scaling choice.
>
> ***A:*** Thank you for pointing this out. We agree that the notation is dense. In the appendix we already provide a detailed explanation of all notations used in the analysis. Following your suggestion, in the revised version we will additionally include a summary table in the main text to make the notation easier to parse and to give clearer intuition for each scaling choice.
>
> >***W3:*** **Lack of Sharpness at $\alpha=2$**. When $\alpha = 2$ (finite variance), the bound leaves a polynomial gap from the optimal O(1/√K) rate. The authors acknowledge this, but it remains an open issue.
>
> ***A:*** Thank you for raising this important point. We agree that in the finite-variance case ($\alpha = 2$) our high-probability bound for the last iterate of clipped SGD exhibits a polynomial gap compared to the optimal $O(1/\sqrt{K})$ expectation rate. We already acknowledge this limitation in the paper. However, this does not diminish the contribution for several reasons.
>
> First, our setting is substantially more challenging: we aim at *high-probability* last-iterate guarantees for *Сlipped*-SGD under a finite-moment noise model. To the best of our knowledge, no such bounds were previously available, even in the basic smooth and convex setting. Importantly, it is not known whether the optimal expectation rate $O(1/\sqrt{K})$ can be upgraded to a high-probability guarantee with only $\mathrm{polylog}(1/\delta)$ dependence on the failure probability $\delta$. In this sense, the gap in our result — our rate scales as $O(K^{-1/3})$ (ignoring additional parameters) — does not contradict any known theory, and closing it remains an open problem.
>
> Second, even in the standard *unclipped* case with finite variance, sharp last-iterate analyses are extremely recent. The optimal last-iterate rate $O(1/\sqrt{K})$ for smooth convex objectives was established only recently (Liu \& Zhou, 2023), while for nearly a decade the state of the art was the $O(K^{-1/3})$ rate of Bach \& Moulines (2014). Likewise, almost-sure last-iterate convergence rates were obtained only in very recent work (Liu & Yuan (2022)). In this context, our result provides, to the best of our knowledge, the first high-probability last-iterate guarantee for Сlipped-SGD under heavy-tailed noise.
>
> For these reasons, we believe that although the bound is not sharp when $\alpha = 2$, it still represents a meaningful advance and opens several promising directions for future work—including the possibility of matching the optimal $O(1/\sqrt{K})$ rate in high probability.
>
> Please refer to the following official comment for the notes on W4 and Q1.

---

> > ### Author Response · Authors · 2025-11-20
> > **Continuation of the response to Reviewer Mpxc**
> >
> > This comment is a continuation of the previous one, and here we respond to W4 and Q1.
> >
> > >***W4:*** **δ-Dependence Obscures Practical Interpretation** The interplay between the learning rate γ and the probability parameter δ—key to your question—is not explicitly analyzed beyond asymptotic scaling.
> >
> > >***Q1:*** In your final convergence result, the learning rate...
> >
> > ***A:***  We respectfully disagree with the statement that this dependence is *quite unusual* in the stochastic optimization literature. In fact, in high-probability analyses, especially under heavy-tailed noise and/or with clipping, it is standard that algorithmic parameters (stepsizes, clipping levels, batch sizes) depend explicitly on the target confidence level $\delta$, typically through factors such as $\log(1/\delta)$ or $\mathrm{polylog}(1/\delta)$. This is the case, for instance, in recent works on clipped or robust SGD under heavy-tailed noise (e.g., Nazin et al., 2019; Gorbunov et al., 2020; Sadiev et al., 2023; Nguyen et al., 2023; Li et al., 2023; Li & Zhou, 2023; Parletta et al., 2025), where the high-probability guarantees are obtained for schedules and thresholds that explicitly depend on $\delta$. Our result is fully aligned with this line of work.
> >
> > We would like to emphasize that the dependence of $\gamma_k$ on $\delta$ is not essential for the validity of our high-probability result, and mild as $\gamma_k \propto \frac{1}{\mathrm{polylog}(1/\delta)}$.
> >
> > Moreover, all our bounds remain true if one uses a stepsize schedule that does *not* depend on $\delta$: in that case, the rate exhibits the standard $\mathrm{polylog}(1/\delta)$ dependence that arises in high-probability analyses via martingale concentration under heavy-tailed noise. The purpose of the $\delta$-dependent tuning in our theorem is much more modest: it improves the polynomial factors inside the $\mathrm{polylog}(1/\delta)$ term. This is a common practice in high-probability stochastic optimization (see, e.g., Nazin et al. 2019; Gorbunov et al. 2020; Sadiev et al. 2023; Nguyen et al. 2023; Li et al. 2023; Li & Zhou 2023; Parletta et al. 2025).
> >
> > Importantly, this choice does *not* affect convergence, does not *artificially stabilize* the dynamics, and is not required for the correctness of the result. It only sharpens the dependence of the final bound on $\delta$ by preventing unnecessary polynomial blow-ups.
> >
> > Indeed, the example mentioned by the reviewer is not related to SGD, Clipped-SGD, or any optimization dynamics considered in our paper. It violates all structural assumptions of our setting (convexity and smoothness). Furthermore, the dynamics described by the reviewer diverges with high probability for any schedule $\gamma_k > 0$ if $x_0 \gg 1$ (if $z_k \equiv 0$, it diverges for any $x_0 > 1$). In arbitrary nonlinear stochastic recursions, shrinking $\gamma_k$ in a pathological way can indeed ''slow down'' divergence on finite horizons, but this phenomenon has no connection to Clipped-SGD. Our stepsizes remain within the admissible regime that lead to convergence (one cannot just slow-down $\gamma_k$ too much, as we need to reach the optimum); they cannot be used to *artificially stabilize* a divergent process or to hide instability of the underlying dynamics.
> >
> > We have clarified this point in the revised version (right before Remark 1) to avoid the impression that the analysis fundamentally relies on a $\delta$-dependent stepsize.

---

> > > ### Comment · Reviewer_Mpxc · 2025-11-20
> > > **reply**
> > >
> > > Thank you for the detailed response. I would like to clarify the core of my concern.
> > >
> > > My question is the following: are you certain that it is possible to derive a high-probability bound with the standard \mathrm{polylog}(1/\delta) dependence without making the stepsize schedule depend on \delta, and without modifying any of your current assumptions—including the assumption that the stochastic gradients only possess moments strictly less than order 2 (e.g., 1+\epsilon moments)?
> > > If such a result is achievable, I believe it would be strictly stronger and conceptually much cleaner than the current form.
> > >
> > > The motivation behind this question is that, in my view, a high-probability convergence guarantee should meaningfully imply the corresponding moment-based (e.g., expectation) guarantee via integrating the tail bound. However, once the stepsize schedule itself depends on \delta, this implication is no longer valid: the high-probability statement then corresponds to a family of algorithms parameterized by \delta, rather than to a single algorithm whose moment behavior one may wish to analyze. As a result, the proposed high-probability guarantee cannot be used to derive any moment-based convergence result for the underlying algorithm.
> > >
> > > This is why I remain concerned that introducing \delta-dependent stepsizes obscures the interpretation and usefulness of the result. If the authors can indeed obtain the same \mathrm{polylog}(1/\delta) dependence under a \delta-independent stepsize schedule while maintaining all existing assumptions (including the <2-moment condition), I would strongly encourage them to present or emphasize this version, as it would address the conceptual issue and considerably strengthen the contribution.

---

> > > > ### Comment · Reviewer_Mpxc · 2025-11-20
> > > >
> > > > In addition, I am aware that several recent works in the literature present high-probability convergence results under stepsizes that explicitly depend on \log(1/\delta). However, I do not believe that this practice should be encouraged. Mathematically, this introduces what is essentially a form of “cheating”: by letting the algorithmic parameters scale with \delta, one can manufacture high-probability stability that does not reflect the actual dynamical behavior of the original system. The example I provided earlier already illustrates this point: even in that setting, it is possible to obtain convergence under stepsizes constrained only by \mathrm{polylog}(1/\delta), without altering the original assumptions. In fact, the argument can be extended to arbitrary noise distributions with exponential tails whose means can be controlled by a \mathrm{poly}(x_n) bound.
> > > >
> > > > The reasoning is standard: one constructs a first hitting time \tau to some set G, takes an arbitrary p/4-th moment of the recursion, applies Burkholder’s inequality to bound the noise term via the quadratic variation, then sums over p/p! to obtain exponential moments, and finally chooses the stepsize smaller than a \mathrm{polylog}(1/\delta) threshold to control the terms involving x_n. A classical Doob-type maximal inequality argument then yields a bound on the probability that the stopping time exceeds T. This entire scheme is well-known from textbook stochastic-process exercises, and—precisely because of that—I do not consider such results publishable in themselves, nor do I think they reflect any intrinsic “convergence” property of the underlying optimization dynamics.
> > > >
> > > > For these reasons, I remain concerned that the introduction of \delta-dependent stepsizes obscures the conceptual meaning and interpretability of the result. If the same \mathrm{polylog}(1/\delta) dependence can indeed be obtained under a \delta-independent stepsize schedule while preserving all of your current assumptions (including the <2-moment condition), I strongly encourage the authors to present or emphasize that version, as it would significantly strengthen the contribution and avoid the conceptual issues outlined above

---

> > > > > ### Author Response · Authors · 2025-11-26
> > > > > **Clarification on $\delta$-dependent parameters and in-expectation guarantees**
> > > > >
> > > > > Dear Reviewer,
> > > > >
> > > > > Thank you very much for your helpful clarifications. We have uploaded the revised version of the paper. Additional changes compared to the previous version are highlighted in magenta.
> > > > >
> > > > > In the new version, we have refined our discussion of the $\delta$-dependence in $\gamma_k$ and $\lambda_k$ and adjusted our statements regarding the potential extension of the current analysis to the case of $\delta$-agnostic hyperparameters (see the text preceding Remark 1).
> > > > >
> > > > > Furthermore, as explained in Section 4.2 of the revised submission, our high-probability convergence results do, in fact, imply a meaningful in-expectation bound. This section now includes a complete derivation and presents the resulting in-expectation guarantee in Corollary 2.
> > > > >
> > > > > We hope that these revisions satisfactorily address your remaining concerns.
> > > > >
> > > > > Best regards,
> > > > >
> > > > > Authors

---

> > > > > > ### Comment · Reviewer_Mpxc · 2025-11-26
> > > > > >
> > > > > > Dear Authors,
> > > > > > Thank you for the detailed clarifications and the revised version of the paper. I have read the new manuscript carefully. The new moment bounds now appear convincing, and the revisions overall improve the clarity and soundness of the work.
> > > > > > I am satisfied with the changes, and accordingly, I have increased my score to 6.
> > > > > >
> > > > > > Best regards,
> > > > > > Reviewer

---

> > > > > > ### Comment · Reviewer_Mpxc · 2025-11-26
> > > > > >
> > > > > > In addition, I would also like to note my appreciation for an additional insightful point the authors highlighted: for clipped SGD, the iterate at time T naturally admits a T-dependent upper bound. This observation indeed implies that meaningful moment bounds can still be obtained even when the learning rate depends on $\delta$. This pointed out to me a useful trick relevant to such algorithms.
> > > > > >
> > > > > > In light of these clarifications and improvements, I have decided to raise my score to **8**.

---

> > > > > > > ### Author Response · Authors · 2025-11-27
> > > > > > >
> > > > > > > Dear Reviewer Mpxc,
> > > > > > >
> > > > > > > Thank you very much for your thoughtful engagement throughout the discussion phase and for appreciating our clarifications and improvements. We are truly grateful for the time and care you devoted to understanding the finer points of our work, and we sincerely appreciate the revised evaluation.
> > > > > > >
> > > > > > > If you find the paper compelling in its current form, we would be very grateful if you could consider championing it during the AC-Reviewers discussion. Your perspective and familiarity with the technical nuances would be invaluable.
> > > > > > >
> > > > > > > Thank you again for your constructive feedback and for helping strengthen the paper.
> > > > > > >
> > > > > > > Best regards,
> > > > > > >
> > > > > > > Authors

---

> > > > > > > > ### Comment · Reviewer_Mpxc · 2025-11-28
> > > > > > > >
> > > > > > > > No problem, I will try my best :).

---

### Meta-Review · Area_Chair_xBKf · 2026-01-17

**Summary:**

This paper aims to solve the problem of the rate of the last iterate of clipped SGD is when the convex objective f (in the problem $\min_x f(x)$) has stochastic gradients with finite $\alpha$-th moment with $\alpha\in(1,2]$. The analysis uses the framework from Taylor & Bach, 2019 who used the performance estimation problem. The resulting rate is $O(1/K^{2(\alpha-1)/3\alpha})$. This seems to be the first rate for the last iterate of clipped SGD with the moment assumption given in the paper. This rate does not match the best-known rates when $\alpha=2$ since it reduces to $O(K^{-1/3})$ compared to the standard $O(K^{-1/2})$ that is obtained by unclipped SGD. Moreover, Liu and Zhou, 2023 also received this faster rate for SGD with sub-Weibull noise.

As the authors also describe, the framework they use for the last iterate guarantee is different from that of Liu and Zhou, 2023 who used Zamani and Glineur's framework. I think that it would have been insightful if the authors told us what happens when they try to integrate clipped SGD in Zamani and Glineur's framework. If the authors tried and this didn't work, then this would make the result of the current paper more interesting. At the moment, the reason for the nonoptimal rate is unclear. On the other hand, all the reviewers agree that the result is extending the state of the art.

Please make your proofs more readable especially the very complicated inequalities after page 21 until the end.

**Reviewer Concerns:**

The main concern pointed out by all the reviewers, which is acknowledged by the authors in their main text, is that the rate is sub optimal. The authors respond by saying that their result is still improving the state-of-the-art which seems to be correct, however, there seems to be no insight on where the looseness in the analysis may be.

The authors resolved the concerns of Reviewer Mpxc relating to the dependence of the algorithmic parameters to the confidence parameter and also obtaining expectation rates.

**Reviewer Scores:**

Reviewer Mpxc would increase their score since their concerns are addressed about the algorithmic parameters and expectation rates.

Reviewer 7EUw would not increase their score since the result they have in mind that would give an optimal rate without projection is not completely clear to the authors.

Reviewer TuTb would keep their rating since they highlighted many drawbacks for the work, including suboptimality of the results and lack of rigor in some estimations.

Reviewer BKzV may have increased their score to a 4 but not to an accept score since their main issue is suboptimality of the result.

---

### Decision · Program_Chairs · 2026-01-26

Accept (Poster)